# Molecular anchoring of free solvents for high-voltage and high-safety lithium metal batteries

Zhuangzhuang Cui[1], Zhuangzhuang Jia[2], Digen Ruan[1], Qingshun Nian [1],
Jiajia Fan[1], Shunqiang Chen[1], Zixu He[1], Dazhuang Wang[1], Jinyu Jiang[1], Jun Ma[1],
Xing Ou [3], Shuhong Jiao [1], Qingsong Wang [2] ✉ & Xiaodi Ren [1] ✉

Constraining the electrochemical reactivity of free solvent molecules is pivotal for developing high-voltage lithium metal batteries, especially for ether solvents with high Li metal compatibility but low oxidation stability ( <4.0 V vs Li$^+$/Li). The typical high concentration electrolyte approach relies on nearly saturated Li$^+$ coordination to ether molecules, which is confronted with severe side reactions under high voltages ( >4.4 V) and extensive exothermic reactions between Li metal and reactive anions. Herein, we propose a molecular anchoring approach to restrict the interfacial reactivity of free ether solvents in diluted electrolytes. The hydrogen-bonding interactions from the anchoring solvent effectively suppress excessive ether side reactions and enhances the stability of nickel rich cathodes at 4.7 V, despite the extremely low Li$^+$/ether molar ratio (1:9) and the absence of typical anion-derived interphase. Furthermore, the exothermic processes under thermal abuse conditions are mitigated due to the reduced reactivity of anions, which effectively postpones the battery thermal runaway.

Lithium (Li) metal is an ideal anode material with an extremely high specific capacity (3860 mAh g$^{-1}$), and the lowest electrochemical potential (−3.04 V vs reversible hydrogen electrode)[1–3]. However, the lack of high-efficiency electrolytes has hindered the development of high-voltage Li metal batteries (>4.0 V, vs Li/Li$^+$, same hereinafter)[4–7]. Conventional carbonate electrolytes are highly reactive to Li anode, and the derived porous and heterogeneous solid-electrolyte interphase (SEI) cannot prevent further side reactions of the electrolyte, which would lead to continuous Li consumption and dendrite growth[2,8–10]. In contrast, ether electrolytes stand out as a promising solution because of their superior reduction stability, lower viscosity, and faster Li$^+$ transport dynamics[11–13]. Unfortunately, the oxidation stability of conventional ether electrolytes (-1 M) is very limited

(<4.0 V), thus severely restricting their application with high-voltage cathodes, especially nickel (Ni)-rich cathodes with highly reactive surface sites (such as Ni$^{3+/4+}$)[14–17].

Recently, extensive efforts have been made to address the compatibility issue between ether-based electrolytes and high-voltage cathodes[18–21]. Apart from artificial cathode-electrolyte interphases (CEIs) that prevent direct contact between high-voltage cathodes and free ether molecules by physical isolation or nanopore-based desolvation[22,23], the fundamental challenge lies in reducing the reactivity of free solvents at the electrode-electrolyte interface. High-concentration electrolytes (HCE) and localized high-concentration electrolytes (LHCE) with high salt/solvent ratios minimize the population of labile-free ether molecules by using the Li$^+$ coordination

[1]Hefei National Research Center for Physical Sciences at the Microscale, CAS Key Laboratory of Materials for Energy Conversion, Department of Materials Science and Engineering, University of Science and Technology of China, Hefei, Anhui 230026, China. [2]State Key Laboratory of Fire Science, University of Science and Technology of China, Hefei, Anhui 230026, China. [3]Engineering Research Center of the Ministry of Education for Advanced Battery Materials, School of Metallurgy and Environment, Central South University, No.932 South Lushan Road, Changsha, Hunan 410083, PR China.
✉e-mail: pinew@ustc.edu.cn; xdren@ustc.edu.cn

strategy. More importantly, the enrichment of anions in the inner solvation sheath promotes the formation of inorganic-rich electrode-electrolyte interphases, which was found to be essential as the kinetic barrier for side reactions between reactive anodes/cathodes and ether-based electrolytes[1,24]. For example, Li fluoride (LiF) has been identified as a crucial component of CEI due to its excellent electron-insulating properties as well as physical and electrochemical stability[25–28]. Despite the encouraging developments of ether-based HCEs or LHCEs, they are still restricted by limited anodic stability (<4.5 V), low ion transport kinetics arising from strong Li⁺-anion interactions, high salt cost, and unsatisfactory low-temperature performance[29,30].

Furthermore, the safety issue of Li metal batteries (LMBs) using concentrated electrolytes with reactive anions has rarely been discussed. Several recent studies have concluded that the safety of electrolytes cannot be justified solely on the basis of their non-flammability[31–33]. Exothermic reactions between the electrolyte and electrode materials are crucial for inducing battery thermal runaway reactions[34,35]. Ouyang and co-workers reported that battery thermal runaway was triggered by the huge amount of heat released by the reaction between the graphite anode and Li salt in concentrated electrolytes, even though a flame-retarding phosphate ester solvent was used[36]. Similarly, Xu and co-workers found that the heat released by the reaction between the electrolyte and the charged cathode (Li iron phosphate after Li⁺ removal) or anode (graphite after Li insertion) is positively correlated with the Li salt concentration[37]. Despite the LHCE reduces the nominal concentration of Li salts (to 1–1.5 M) with the addition of diluents, the temperature of the battery rises sharply during the puncture test to damage the battery structure due to the enhanced reactivity of anions in LHCE. The reaction between Li salts and Li metal anode in concentrated electrolytes could potentially release a large amount of heat to induce the thermal runaway avalanche reaction due to the low melting point (180 °C) and highly reactive nature of Li metal. Therefore, it is urgent to develop effective strategies to design ether-based electrolytes with high Li metal Coulombic efficiency (CE), high-voltage stability, and, last but not least, suppressed exothermic reactions with Li metal to improve the safety of LMBs.

Herein, we propose a molecular anchoring diluent electrolyte (MADE, down to 0.19 M) with a wide electrochemical stability window (>4.7 V on NMC811 cathode) and greatly increased thermal runaway temperature for LMBs. In contrast to the widely-used approach of restricting free solvents through Li⁺ coordination in concentrated electrolytes, we exploit the strong non-conventional hydrogen bond interactions with oxidation-resistant hydrofluoroether and ether molecules to improve the oxidation resistivity of ether solvents. The MADE design with decreased anion reactivity not only demonstrates enhanced oxidation stability (>4.7 V) compared to LHCE, which rely on extensive anion sacrificial decomposition, but also fosters a more flexible SEI layer to accommodate volume changes of Li metal during cycling. Furthermore, the thermal runaway temperature of LMBs is greatly increased (from 141 to 209 °C) due to suppressed exothermic side reactions with Li metal. This work sheds light on a promising approach to expanding the electrochemical stability window and enhancing the safety of electrolytes for LMB applications.

## Results and discussion
### The dilute electrolyte design with molecular anchoring
To investigate the impact of hydrofluoroether molecular "anchor" (e.g., 1,1,2,2-tetrafluoroethyl-2,2,3,3-tetrafluoropropylether, TTE) on the electrochemical stability of the ether solvent (1,2-dimethoxyethane, DME), we formulated the electrolytes with a high TTE to DME molar ratio of 3:1, while the ratio of the Li bis(fluorosulfonyl)imide (LiFSI) salt was kept low to minimize the ion-solvent interactions. Therefore, electrolytes with low molar ratios of LiFSI: DME: TTE, 1:9:27 (~0.19 M), 2:9:27(~0.38 M), and 3:9:27 (~0.56 M) were prepared and

labeled as MADE-1, MADE-2, and MADE-3, respectively. To gain insight into the thermodynamic interaction between DME and TTE molecules, the heat of mixing was measured using isothermal titration calorimetry (ITC). Upon the addition of DME into TTE, distinct exothermic peaks were detected, with each drop estimated to release 706.1 cal mo1⁻¹ of heat (Fig. 1a and Supplementary Fig. 1). By contrast, significantly reduced heat release (185.9 cal mol⁻¹) was observed when HCE (LiFSI-1.2DME in molar ratio) was added into TTE. This suggests that the DME-TTE interaction is greatly eliminated when most DME molecules are strongly confined in the inner solvation sheath (Li⁺-O_(DME) coordination).

Meanwhile, the dominant interaction between DME and TTE in MADEs is not from the pseudo hydrogen bond between H_(DME) and F_(TTE). As shown in Fig. 1b, despite the ¹H-¹⁹F coupling signals observed in two-dimensional nuclear magnetic resonance (NMR), the pseudo H bonds between C-H (DME)...F-C (TTE) are weaker than those between H_(TTE) and O_(DME)[38,39]. Further ¹H NMR results indicate the H bond between O_(DME) and H_(TTE) accounts for the apparent heat release observed in the ITC (Supplementary Fig. 2). When DME was mixed with TTE, the chemical shifts of H_(TTE) increased while those of H_(DME) decreased, indicating that H_(TTE) acts as the H bond donor with O_(DME) as the acceptor[40]. Although the spatial proximity of H_(DME) and F_(TTE) induces relatively weak interactions, which would potentially cause the ¹H chemical shift of DME to a higher value (H_(DME) as the H bond donor instead), the dominant force is the H bond between H_(TTE) and O_(DME) and the resulting ¹H chemical shift of DME is toward a lower value. Further geometry optimizations of DME-TTE complexes by density-functional theory (DFT) also suggest that the H_(TTE)...O_(DME) interactions are favored compared to H_(DME)...F_(TTE) interactions (Supplementary Fig. 3), which is consistent with previous studies[39].

To further verify the hydrogen-bonding between TTE and ether molecules, we carried out Fourier-transform infrared spectroscopy (FT-IR) analysis in CCl₄ solutions. To study the effect of hydrogen-bonding on the C-H vibration in TTE while avoiding the interference of C-H bonds in ether (overlapping signals), deuterated tetrahydrofuran (THF-$d_8$) was selected because of its characteristic ethereal moiety and easy accessibility. As shown in Fig. 1c, after adding THF-$d_8$ to the TTE solution, the CF₂-H vibration peak shifts noticeably from 3000 to 3010 cm⁻¹, which aligns with the featured blue-shifting vibration of CF₂-H after H bond formation reported in the literature[38,41,42]. In addition, we study the interaction between TTE and DMSO, which is often selected to study the C-H...O hydrogen bond as it contains no hydrogen donor[43]. Similarly, upon adding DMSO-$d_6$, a noticeable blue shift of the CF₂-H signal in TTE was also observed, which can demonstrate the hydrogen bond donating ability of TTE. Moreover, with the content of TTE increasing, the ¹⁷O-NMR signal of DME shifts to a lower chemical shift (Supplementary Fig. 4), which agrees with the previous study that hydrogen-bonding induces an upfield chemical shift of oxygen atoms[44]. In contrast, there is no apparent shift of the ¹⁷O-NMR signal for the oxygen atom in TTE. These findings support the conclusion that TTE forms hydrogen bonds with DME through interactions between the CF₂-H_(TTE) and O_(DME).

Furthermore, DFT calculations were employed to obtain more detailed coordination configurations (Supplementary Fig. 5a). From the deformation electron density, the interaction for DME-TTE, or DME-DME complexes could be identified with the overlapping deformation electron density, while the interaction between the C-H_(TTE) and C-F_(TTE) is not observed on the 0.03 a.u. isovalue map for the TTE-TTE complex. Meanwhile, as shown in the electrostatic potential mapping (ESP) (Supplementary Fig. 5b), the minimum electrostatic potential energy in the vicinity of oxygen of DME increases notably after its coordination with TTE (from −42.75 to −17.22 kcal mol⁻¹), which has substantial effects on the oxidative stability of the molecules[45].

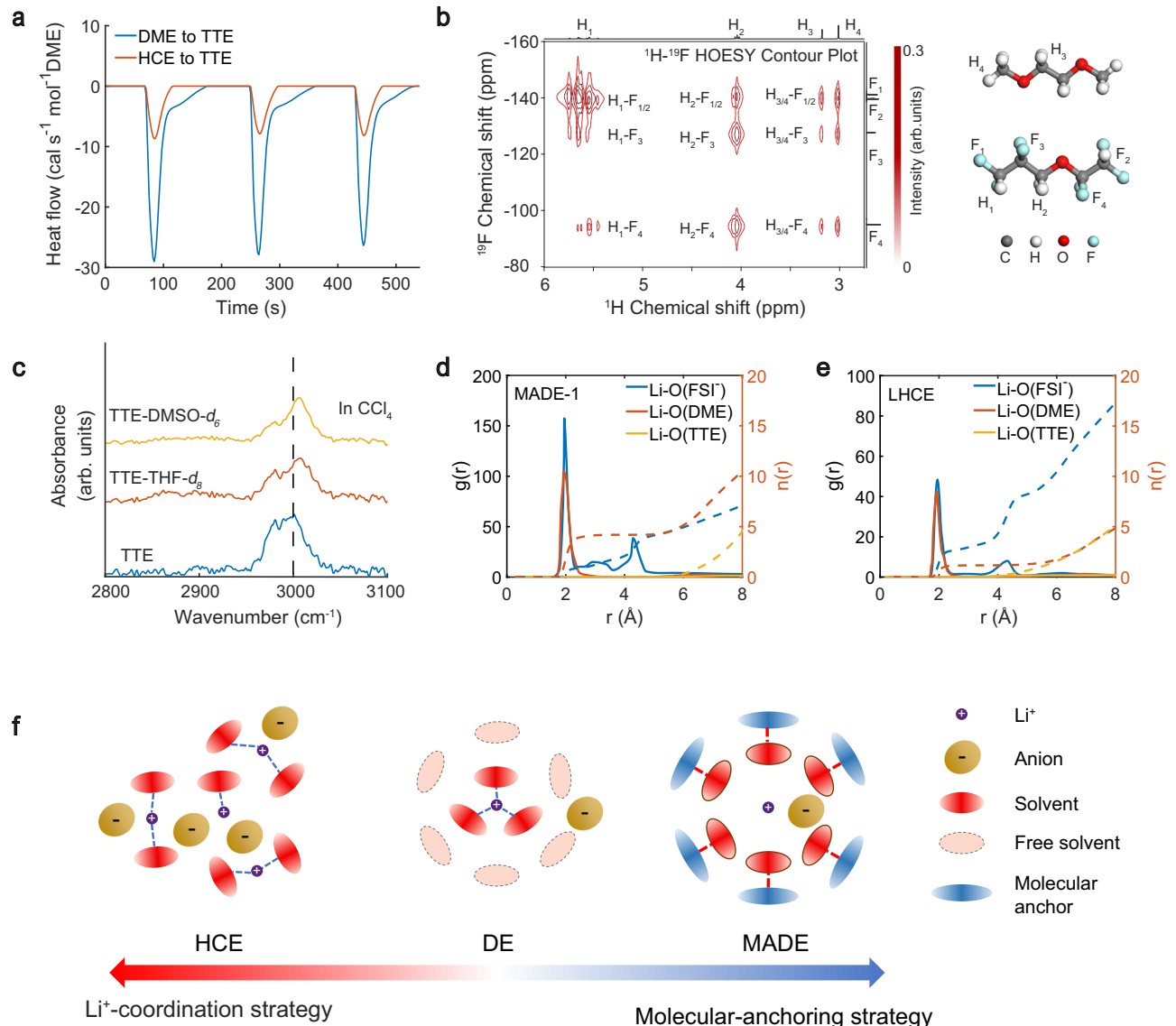

**Fig. 1 | Molecular interactions and electrolyte design strategy. a** Heat flow curves with DME and HCE added into TTE. **b** The two-dimensional $^1$H-$^{19}$F Heteronuclear Overhauser Effect Spectroscopy (HOESY) for DME-TTE (1:3 in molar ratio). **c** Comparison of C-H vibration signals of TTE before and after mixing with THF-$d_8$ and DMSO-$d_6$ in CCl$_4$. **d, e** The radical distribution functions and coordination numbers calculated from MD simulations of MADE-1 and LHCE. **f** Schematic diagram of MADE and HCE/LHCE design strategy (DE, LiFSI-9DME in molar ratio).

The evolution of Li$^+$ coordination structures from MADE-1 to LHCE (LiFSI-1.2DME-3TTE in molar ratio) was investigated using Raman spectroscopy (Supplementary Fig. 6). The region between 700 and 760 cm$^{-1}$ corresponds to the S-N-S bond stretching vibration of the FSI$^-$ anion[46,47]. The two Raman peaks of pure DME at 821 and 848 cm$^{-1}$ are C-O stretching and CH$_2$ rocking signals[48,49]. The vibration signal of free FSI$^-$ anions (720 cm$^{-1}$) in the dilute electrolyte (DE, LiFSI-9DME in molar ratio) could easily be identified, while the anions exist in the form of contact ion pairs (CIP, 730 cm$^{-1}$) in MADEs. This is because the coordination of TTE weakens the binding affinity between DME and Li$^+$. As the salt concentration increases, the signals associated with Li$^+$-coordinated DME (877 cm$^{-1}$, C-O stretching) and ion aggregates (AGG, 753 cm$^{-1}$, S-N-S stretching) gradually intensify[50,51]. Similar changes in the IR spectrum are observed as the new peak associated with Li$^+$-coordinating DME at 1079 cm$^{-1}$ becomes stronger with the increasing salt content[52] (Supplementary Fig. 7), which agrees with our argument that the dominant species gradually transform from DME-TTE complexes in MADEs into Li$^+$-DME complexes in LHCE. To further quantify the differences in anionic coordination. The solvation structures of

MADEs and LHCE were investigated further using molecular dynamic (MD) simulations, and the final snapshots of the structures are shown in Supplementary Fig. 8. Based on the results of the Radical distribution functions (RDF) presented in Fig. 1d, e, the O atoms of both DME and FSI$^-$ anions dominate the primary coordination shell of Li$^+$ (within 2.6 Å), while the coordinated O atoms of TTE are insignificant. Moreover, the coordination number of O$_{(DME)}$ is 3.7 times higher than that of O$_{(FSI^-)}$ in MADE-1, and this ratio gradually decreases to 1.31 for MADE-3 and 0.42 for LHCE, suggesting that the dominant interactions gradually shift to electrostatic interactions of ionic aggregates. Based on the aforementioned discussion, the difference between MADE, DE, and HCE or LHCE was shown in Fig. 1f. In contrast to the strategy of restricting free ether molecules with Li$^+$-coordination, the MADE design employs a new molecular anchoring approach to suppress the reactivity of free solvents.

## The enhanced electrolyte oxidation stability
Electrochemical test results indicate that significantly enhanced oxidation stability of ether electrolytes can be achieved in the electrolytes

with MADE solvation structure. Alumina (Al) foil coated with PVDF-Super P carbon nanoparticles was used as a high-surface-area working electrode to evaluate the onset potential of the MADEs. As shown in Fig. 2a and Supplementary Fig. 9, an interesting trend of higher oxidation onset potentials was found as the salt ratio decreased, where MADE-1, MADE-2, and MADE-3 are stable beyond 4.7 V (highest at ~4.77 V for MADE-1), while the oxidation onset potential for the LHCE was around 4.60 V. In addition, Li‖NMC811 full cells were assembled to verify the electrochemical stability of the MADEs on reactive cathodes under high voltages. As shown in Supplementary Fig. 10, highly reversible charge-discharge cycling could be achieved in MADE-1, MADE-2, and MADE-3 at 4.7 V cut-off voltage, while the electrolytes with higher salt concentrations all suffered serious overcharging in the first two cycles. The leakage current tests at 4.6 and 4.7 V (Fig. 2c) also confirmed that the MADEs have improved high-voltage stability compared to that of the LHCE. It is of paramount importance to note that this stabilization mechanism exhibits a broad generality, with comparable phenomena manifesting across diverse solvents (other ether solvents or diluents, e.g., diglyme (G2), 1,1,2,2-tetrafluoroethy-2,2,2-trifluoroethyl ether (HFE); or organic carbonate solvents, e.g., dimethyl

carbonate (DMC); or salts, e.g., Li hexafluorophosphate (LiPF$_6$), Li bis-trifluoromethanesulfonly imide (LiTFSI)) (Fig. 2b). On the other hand, augmenting the concentration of anchoring agents proves advantageous in enhancing the oxidation stability of the electrolyte, as sufficient anchoring mitigates the reactivity of free DME at the interface (Supplementary Fig. 11). Moreover, the Al corrosion current remains below 2 μA cm$^{-2}$ in MADEs and LHCEs utilizing LiFSI or LiTFSI due to the passivation effect of TTE[20] and no apparent Al corrosion could be observed from SEM images after CV scans (Supplementary Fig. 12).

Li‖NMC811 cells were further evaluated by cyclic voltammetry and galvanostatic cycling tests. As shown in Fig. 2d, e, the MADE-1 demonstrated a greatly suppressed oxidation reaction beyond 4.6 V during CV scans, which is in stark contrast to the LHCE and the DE (Supplementary Fig. 13). Over the subsequent three cycles, the peaks in MADE-1 exhibited a notable low polarization, while the cells employing DE or LHCE showed distinct escalations in polarization. From the cycling performance at 0.33 C (1 C = 200 mAh g$^{-1}$), the cell using LHCE displays fast capacity decay along with an enlarged charging and discharging voltage polarization, which are indicative of the deterioration of the cathode due to side reactions and byproduct accumulation

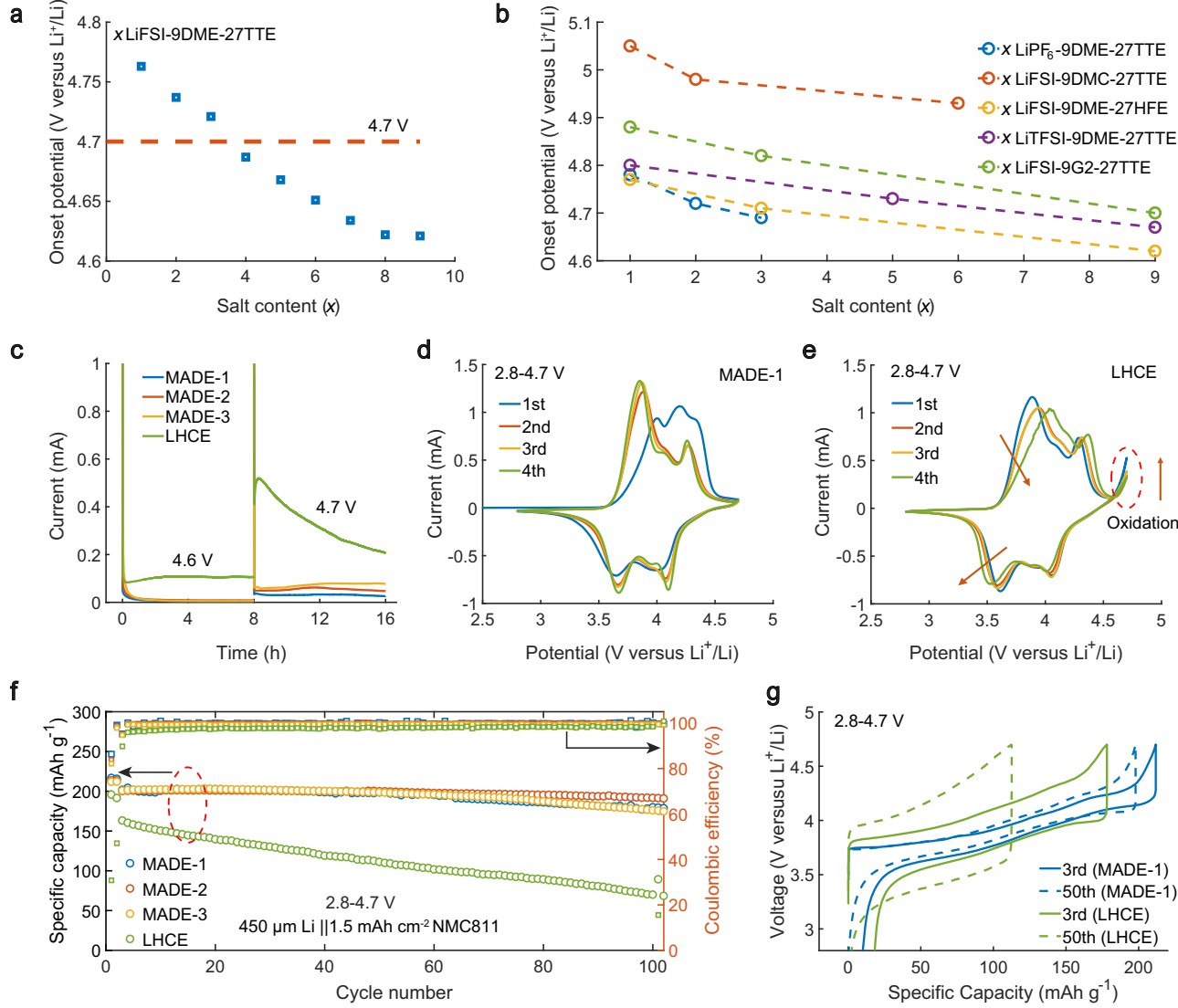

**Fig. 2 | The electrolyte oxidation stability and cell performance. a** The onset potential change with different salt contents in the LiFSI-DME-TTE system. **b** The summary of oxidation potentials with different salts, solvents, and anchoring agents. **c** The leakage currents of Li‖NMC811 cells at 4.6 and 4.7 V. **d, e** CV curves using MADE–1 and LHCE at 0.1 mV s$^{-1}$ within the voltage range between 2.8 and 4.7 V. **f, g** Cycling performance and voltage profiles of Li‖NMC811 cells using MADE-1, MADE-2, MADE-3 and LHCE.

(Fig. 2f, g). In contrast, the cells utilizing MADEs demonstrate substantial improvements in cycling stability and voltage fade (Supplementary Fig. 14). Specifically, the cell with MADE-2 exhibits excellent cycling stability, with a capacity retention of 95.0% over 100 cycles, whereas MADE-1 and MADE-3 retain 88.5 and 86.2% of the initial discharge capacity, respectively. These results exhibit good reproducibility, as evidenced by the repeated data in Supplementary Fig. 15. In addition, compared to LHCE with featured Li$^+$-anion interactions, the MADE design further decreases the electrolyte viscosity and improves the Li$^+$ conductivity (Supplementary Fig. 16). This is highly beneficial for the battery rate capability. Greatly improved discharge capacities at 2 C and 4 C rates could be realized in MADEs than those in LHCE (Supplementary Fig. 17). The relatively lower capacity in MADE-1 of the MADE family is likely due to its extremely low ion concentration. Additionally, apart from TTE, we also found other hydrofluoroethers, such as 1H,1H,5H-Perfluoropentyl-1,1,2,2-tetrafluoroethylether (OTE) and ethyl 1,1,2,2-tetrafluoroethylether (ETE) with the -CF$_2$H moiety, have similar anchoring effect (Supplementary Figs. 18, 19).

## The cathode/electrolyte interface

To elucidate the underlying mechanism of the improved stability of NMC811 cathodes in the MADEs, we characterized the cycled cathodes using scanning electron microscopy (SEM) and high-resolution tunneling electron microscope (HR-TEM). Noticeable differences in surface morphology were observed on cathodes cycled in MADE-1, MADE-2, MADE-3, and LHCE (Fig. 3a–c and Supplementary Fig. 20). Specifically, cathode particles cycled in MADE-1 exhibited a very smooth surface, similar to the pristine cathode. With increasing salt concentrations, different surface features were observed: a rough surface with gradually enhanced visible deposit particles (for MADE-2 and MADE-3) and a relatively smooth surface with some wrinkles (for LHCE). These results highlight an interesting phenomenon whereby higher salt concentrations appear to result in more pronounced side reactions on the cathode under high voltage. This observation may also be related to the darker color of the separator side facing the cathode after cycling in the LHCE, in comparison to those MADEs. This was simultaneously confirmed by HR-TEM images presented in Fig. 3d–f. The CEI layers formed in the MADE-1 and MADE-3 electrolytes were thin and uniform around 1 and 3.2 nm, respectively. Nevertheless, excessive growths of interphase layers on NMC811 cathodes were found after cycling in the LHCE (~20.1 nm). The thickened interface layer was also replicated in Raman mapping experiments (Supplementary Fig. 21). The integration signal of MADE-1 corresponding to NMC811 in the range of 385–700 cm$^{-1}$ is significantly stronger than that of LHCE[53].

The CEI composition of cycled NMC811 cathodes was further identified using X-ray photoelectron spectroscopy (XPS) with Ar$^+$ depth profiling (Supplementary Fig. 22). The CEI layers formed in MADE-1 and MADE-3 were found to be much thinner than that of LHCE, as indicated by the particularly strong M-O signal (531.0 eV, reference: C 1s at 284.8 eV, Supplementary Fig. 23) after initial sputtering (~5 nm). In contrast, the M-O signal underneath the thick CEI layer could be identified for the LHCE cathode sample only after a long time of Ar$^+$ sputtering (Fig. 4b). This agrees with the HR-TEM and Raman mapping results. The C-F (687.6 eV) and LiF (~685.0 eV) signals in the F 1s spectra for MADEs are likely from the decomposition of TTE molecules on the high-voltage cathode surface (Fig. 4a). This was also confirmed by the higher F/S atomic ratios during depth profiling in MADE-1, MADE-3, which indicate the fluorine species come from TTE and the anion reactivity decreases with decreasing salt ratios (Supplementary Fig. 24). In addition, the N 1s and S 2p spectra show clear differences of salt anion reactivity on the cathode surface. In the MADE-3 and the LHCE with higher salt concentrations, we observed more accumulation of the N-O$_x$/S-O$_x$ species, especially for the LHCE, while no apparent such signals were observed in MADE-1 (Fig. 4c and Supplementary Fig. 25).

The dramatic changes of the CEIs in different electrolytes could be verified using X-ray absorption spectroscopy (XAS) in the total electron yield (TEY) mode (probing depth ~5 nm). The characteristic energy absorption of oxygen in the pristine NMC811 material (532.8 and 536.6 eV) were also detected on the electrode after cycling using MADE-1, supporting the formation of a thin cathode-electrolyte interphase layer (Fig. 4d). The energy absorption peaks observed at 539.6 and 541.2 eV on the LHCE electrode could potentially arise from partially oxidized LiFSI, owing to their proximity to the main peak of Li$_2$SO$_4$. This observation is consistent with the results obtained from S L-edge spectra (Fig. 4e). Moreover, the weaker S signal detected on the MADE-1 electrode suggests a lower sulfur concentration at the electrode-electrolyte interface, similar to the XPS results. The ex-situ NMR was further used to prove anion decomposition behavior in LHCE. Supplementary Fig. 26 shows the $^{19}$F spectra of extracted electrolytes from the cells with LHCE before and after 4, 6, and 8 cycles at 0.1 C. As the number of cycles increased, a new signal was detected near the FSI$^-$ peak (53–53.5 ppm) and exhibited continuous enhancement, indicating that the anion was directly involved in the oxidation progress.

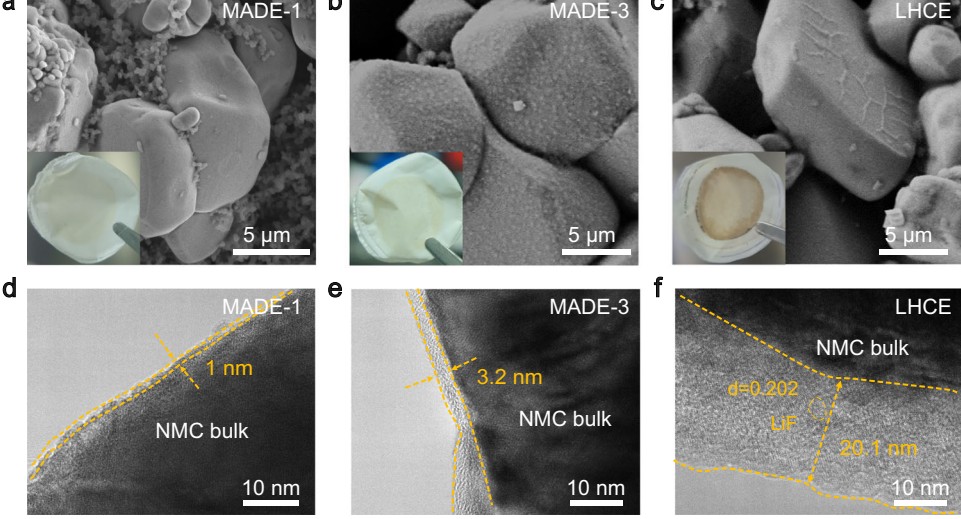

**Fig. 3 | Characterizations of the CEI morphology. a–c** SEM morphologies of the cathodes in MADE-1 (**a**), MADE-3 (**b**), and LHCE (**c**) after 50 cycles. (Insert: optical images of separators facing the cathodes). **d–f** HR-TEM images of the cathodes after 50 cycles in MADE−1 (**d**), MADE-3 (**e**), and LHCE (**f**).

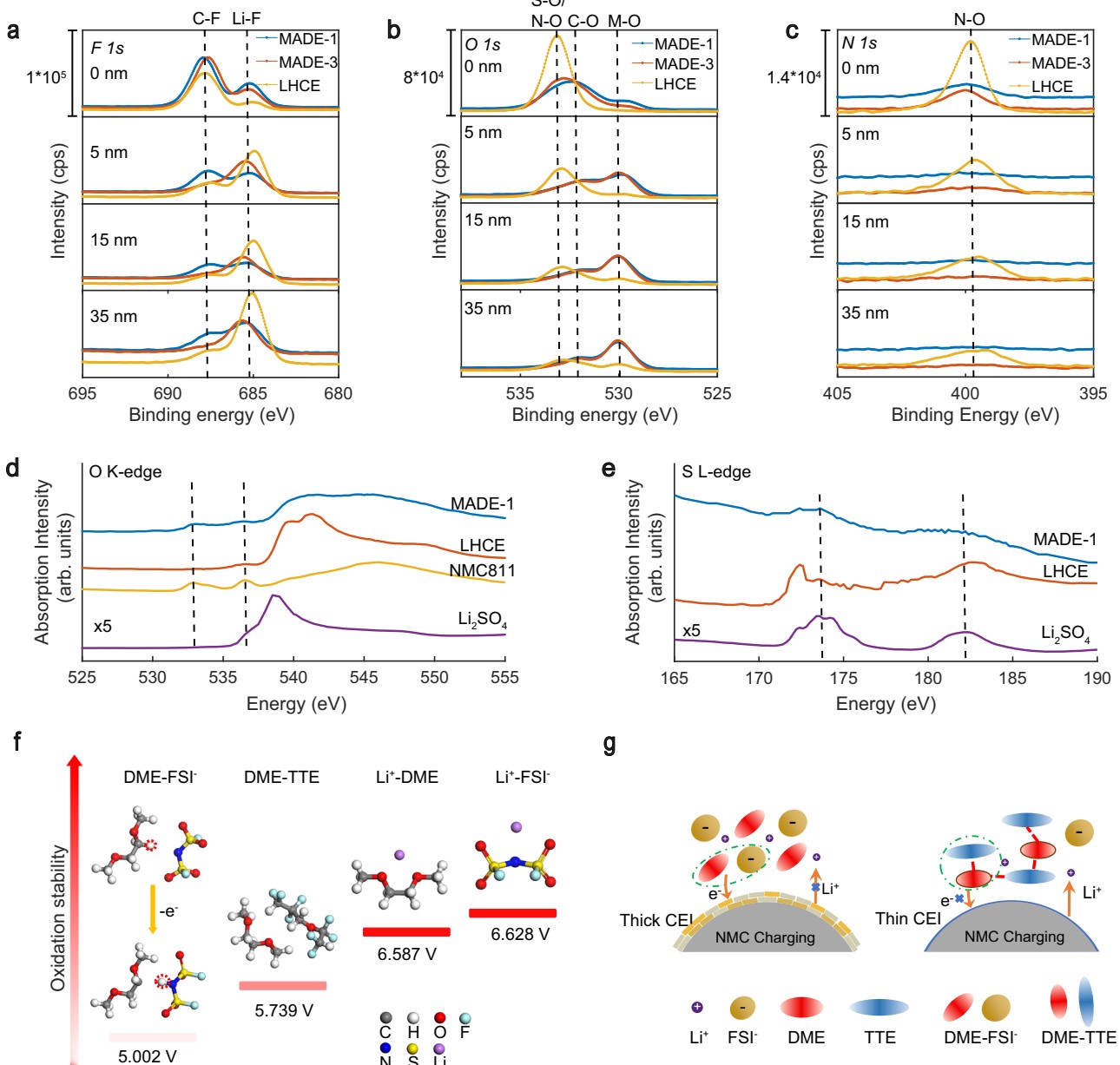

**Fig. 4 | CEI compositions and electrolyte oxidation mechanism. a–c** XPS etching results of the CEIs on NMC811 cathodes after 50 cycles in MADE−1, MADE-3, and LHCE with F 1*s*, O 1*s*, and N 1*s* spectra. **d, e** The XAS spectra of O *K*-edge and S *L*-edge for the cathodes. **f** The calculated oxidation potential of listed complexes. **g** Schematic illustration of electrolyte oxidation behavior during the charging process.

To understand the oxidation reaction mechanisms of the electrolyte, we retrieved the representative configurations of preferred complexes from the MD and further calculated their oxidation potentials with DFT based on the reported calculation method[15]. As shown in Fig. 4f, DME-FSI⁻ complex presents the lowest oxidation potential, companying with the spontaneous H transfer from DME to FSI⁻. Two different configurations from MD were selected for the FSI⁻-DME complexes to get a more generalized result, one with the -CH$_{3(DME)}$ near the N$_{(FSI⁻)}$ site and the other with -CH$_{2(DME)}$ near the N$_{(FSI⁻)}$ site. The final results showed that in both cases, similar oxidation potentials of 5.070 and 5.002 V were obtained (see Supplementary Fig. 27 for calculation details). In contrast, the DME-TTE complex exhibits a much higher oxidation potential (5.739 V). Although the controlled oxidation of DME-FSI⁻ complexes under low voltages (<4.5 V) induces an inorganic-rich CEI to mitigate further electrolyte oxidation[54], increasing the charging cut-off voltage would result in

uncontrollable electrolyte side reactions, thus thickening the CEI and deteriorating the cathode structure. By replacing the dominating electrolyte species from DME-FSI⁻ complexes to H bond anchored DME-TTE complexes, the electrolyte side reactions could be greatly suppressed (Fig. 4g).

It should be noted that the improved oxidation stability of ether molecules in MADEs mainly results from the shift of oxidation potential due to the H-bonding with anchoring agents, rather than the protection by the CEI formed. To further verify the role of the CEI, the cathode recovered from the cell with MADE-1 after three formation cycles at 0.1 C was further tested with LHCE. The appearance of anodic currents beyond 4.6 V with LHCE during the CV test suggests that the suppression of oxidation side reactions should be attributed to the enhanced thermodynamic stability of ether molecules (Supplementary Fig. 28). The improved oxidation stability of MADEs without relying on the protective CEI is important for the reversible structural

transformation of the cathode under high voltages. As indicated by the in situ XRD tests (Supplementary Fig. 29), compared to MADE-1, the cathode recovered from LHCE exhibited smaller layer spacing expansion during the phase transition from H1 to H2 (0.24° for LHCE, and 0.34° for MADE-1), which can be attributed to the large amounts of $Ni^{2+}$ that remain in the Li layer, weakening the repulsion between the oxygen layers, as evidenced by the lower (003)/(104) ratio[55]. The transition from H2 to H3 is ascribed to the dismissed supporting between the transition metal layers caused by the withdrawal of Li ions from the structure. During this phase transition, the volume contraction of the two electrodes is similar, but the charging capacity of LHCE is more limited, likely due to the thick CEI with high impedance[56]. Furthermore, the lower discharge capacity of LHCE can be assigned to a shorter residence time in the H2 phase and rapidly increasing polarization, resulting from the irreversible transformation from the H3 phase to the rock salt phase[57]. In summary, the severe interfacial side reactions not only result in the build-up of byproducts but also cause a series of damage to the layered structure of the cathode.

## The Li metal anode compatibility

The compatibility of the electrolytes with Li metal was tested using Li||Cu half-cells at a current density of 0.5 mA cm$^{-2}$ and a capacity of 1.0 mAh cm$^{-2}$ with a cut-off potential of 1.0 V (Fig. 5a). The Coulombic efficiency with DE fluctuates violently, averaging 87.8% over the first 40 cycles. This is because of the uncontrollable electrolyte side reaction and Li dendrite growth, as evidenced by the SEM morphology of deposited Li (Supplementary Fig. 30). In sharp contrast, large and compact granular particles were formed in MADE-1, where highly efficient Li plating/stripping was achieved with an average CE >99.00%. Increasing the salt ratio to MADE-2 (99.12%) and MADE-3 (99.30%) results in further improvements in Li CE, indicating that both TTE and FSI$^-$ anions are advantageous for Li metal protection. TTE functions as the anchoring agent for DME molecules, weakening the ability of DME to coordinate Li$^+$ and amplifying the proportion of FSI$^-$ in the solvation sheath. Moreover, TTE participates in the formation of the SEI, resulting in an interface layer enriched with fluorinated species, which is beneficial for Li metal protection. However, the LHCE with nearly saturated salt content did not exhibit further increased Li CE (99.24%),

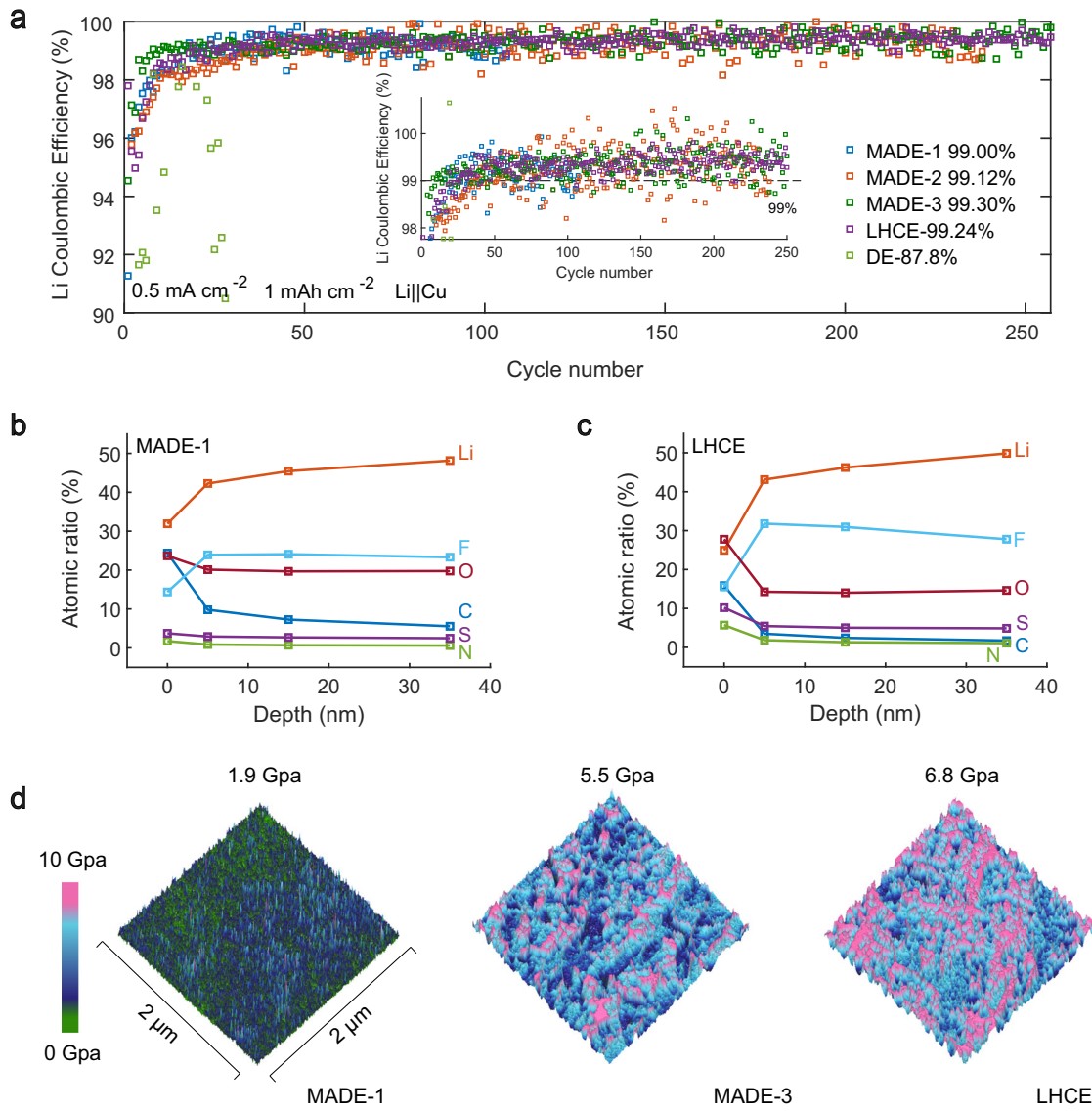

**Fig. 5 | The electrochemical stability of Li metal anode. a** Li Coulombic efficiency in Li||Cu cells using different electrolytes with a current density of 0.5 mA cm$^{-2}$ and area capacity of 1 mAh cm$^{-2}$ (Insert: enlarged view of Li Coulombic efficiency). **b, c** Atomic ratios of SEI from Ar$^+$ sputtering on cycled Li metal anodes in MADE−1 and LHCE. **d** AFM surface modulus for deposited Li metal in MADE−1, MADE−3, and LHCE.

and the overpotential of Li||Cu cell was larger in LHCE (Supplementary Fig. 31). The Li CE results from Aurbach's method exhibit a similar trend (Supplementary Fig. 32). In LHCE, the clustering of anions introduced by a high salt content aggravate the consumption of Li metal by reactions with anion, resulting in only a minor change in the Li CE compared to MADE-3[58] (CV curves shown in Supplementary Fig. 33). Moreover, Li symmetrical cells were tested to evaluate the compatibility of the electrolyte with lithium metal. As depicted in Supplementary Fig. 34, when compared to LHCE, symmetrical cells employing MADEs exhibit a notable reduction in overpotential. Nevertheless, the cycling lifespan is observed as LHCE >MADE-3 >MADE-1, primarily attributable to the lower anion concentration in MADEs. We expect this problem could be mitigated by manipulating the reactivity of the salt used in the future.

To understand the changes in Li CEs, the SEI components formed in MADE-1 and LHCE were further studied by XPS analysis with Ar+ sputtering. The selected elemental contents at various etching depths are presented in Fig. 5b, c. For MADE-1, the atomic ratio of the depth profile displayed a rapidly increasing Li content, confirming the presence of a thin SEI. Additionally, compared to LHCE, the higher ratio of F/S at all sputtering depths suggests enhanced participation of TTE in the SEI construction (Supplementary Fig. 35). On the other hand, clear higher organic C species including C-C/C-H (284.8 eV) and C-F (290.2 eV) are found in the SEI for MADE-1, which enables the modulation of interfacial flexibility and facilitates the suppression of volume expansion associated with plating and stripping (Supplementary Fig. 36). Fourier-transform infrared (FT-IR) spectra of SEI layers also show stronger C–H signals in MADE-1 compared to MADE-3 and LHCE (Supplementary Fig. 37). The enhanced F, N, and S signals detected in LHCE supports the increased participation of FSI⁻ in the formation of the SEI. Furthermore, stronger Ni 2p signals in LHCE suggest more severe cathode corrosion at high voltage and the crosstalk effect on the SEI formation (Supplementary Fig. 38). The mechanical properties of the SEI layers could be further revealed by atomic force microscopy (AFM). Although SEI with a high modulus is advantageous for suppressing Li dendrite growth, it is imperative to realize that the substantial Li volume fluctuations during cycling inevitably result in the disruption and subsequent reformation of the SEI[59,60]. In contrast to the LHCE, the solvation structure in MADEs shows a diminished proportion of anions. This modification serves to balance the decomposition of anions and TTE, thereby promoting the development of a flexible SEI, ultimately reducing the susceptibility of the SEI fragmentation (Fig. 5d). The changes of Li CEs with the LiFSI content indicates that a controllable degree of FSI⁻ decomposition during the SEI formation process in MADEs is preferred, which could promote the formation of a more ionic conductive SEI to improve the uniformity of Li deposition and stripping. However, excessive FSI⁻ decomposition in the LHCE yields a highly rigid SEI layer, which has an adverse effect on the Li CE. Similarly, slight FSI⁻ decomposition on the cathode could help shield the reactive electrode surface from labile "free" solvent molecules generated from dynamic exchanges in the electrolyte, as supported by the improved cathode stability in MADE-2.

## The interfacial reactivity under thermal abuse conditions

The electrolyte solvation structure is also closely related to their exothermic reactions with electrode materials under thermal abuse conditions, which is of paramount importance in battery safety properties. The thermal reactions of electrolytes were first evaluated using differential scanning calorimetry (DSC) with Li anode and charged NMC811 cathode (4.6 V cut-off), respectively. Briefly, a hole was drilled in the cap of the sample holder to prevent the rapture of the holder and potential damage to the instrument due to excessive pressure built up during the heating process. To minimize the influence of electrolyte loss due to thermal evaporation, a thin stainless-steel foil with gold coating was placed underneath the cap, which seals the electrolyte

inside but would break to release excessive pressure when generating too much gas. For Li metal in LHCE, apparent heat release was observed when the temperature rose above ~50 °C. It is likely due to the high reactivity of anions in the LHCE solvation structure (Fig. 6a). As the salt ratio decreases, the exothermic reaction between the electrolyte and Li metal apparently shifts towards higher temperatures, especially for MADE-1, which shows no apparent heat release before the melting point of Li metal (180 °C). This result also correlates with the lower reduction currents of MADE-1 for the SEI formation process in CV scans (Supplementary Fig. 39). Compared to Li metal, the reactions between the electrolytes and charged cathodes are more complex. Similarly, as the salt ratio decreases, the exothermic reactions between electrolytes and charged cathode were significantly suppressed (Fig. 6b). This agrees with the oxidation susceptivity of FSI⁻-DME complexes on delithiated cathodes, as discussed earlier. Moreover, the DE is highly flammable and has a very large self-extinguishing time (SET) of 81 s g⁻¹ (Supplementary Movie 1). In contrast, both the MADEs and LHCE cannot be easily ignited due to the large contents of the fluoroether diluent (Supplementary Fig. 40 and Supplementary Movies 2–4). As a proof-of-concept demonstration, we further compared the thermal runaway process of Li||NMC811 full pouch (Supplementary Table 1) cells employing LHCE and MADE-1 electrolytes (Fig. 6c), particularly around the temperature region of Li melting, which could potentially induce severe safety hazards due to internal short-circuiting or battery structural damage. The pouch cells were first charged at 0.1 C to 4.6 V before being transferred to the accelerating rate calorimeter (ARC). No apparent heat releases were recorded below 100 °C for both MADE-1 and LHCE, which suggests the inorganic-rich SEI layer derived from anion in LHCE could be beneficial for suppressing further reactions with the electrolyte under low temperatures. However, as the chamber temperature reaches 141 °C, the heat generation rate dT/dt exceeds

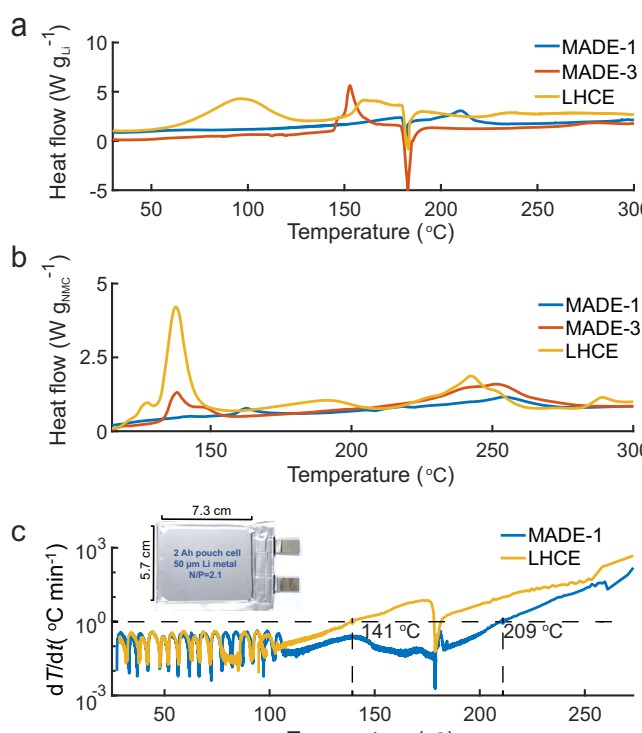

**Fig. 6 | Thermal stability of cell components and pouch cells with different electrolytes. a** Normalized DSC traces for Li metal with MADE−1, MADE-3, and LHCE (Li metal mass: 2 mg). **b** Normalized DSC traces for charged NMC811 cathode with MADE−1, MADE-3, and LHCE (NMC mass: 4 mg). **c** The temperature dependence (dT/dt) of the Li||NMC811 pouch cells with the MADE−1 and LHCE during the thermal runaway process (Insert: the size and optical image of the pouch cell).

1 °C min$^{-1}$, and the battery temperature undergoes an exponential escalation subsequently, which marks the threshold temperature for thermal runaway. In contrast, no apparent heat is released in MADE-1 before Li melts and the thermal runaway temperature is postponed to 209 °C (dT/dt >1 °C min$^{-1}$). Therefore, minimizing the electrolyte reactivity with highly active electrodes by the MADE design offers a promising strategy for the development of next-generation high-energy-density and high-safety LMBs.

In this work, we propose a molecular anchoring approach to restrict the interfacial reactivity of free solvents in dilute electrolytes. The inherent electrochemical characteristics of the electrolyte can be fundamentally altered through the intermolecular H bond, which overcomes the long-standing voltage limitation for dilute ether-based electrolytes. In contrast to HCE or LHCE, which rely on anion decomposition for interface passivation, the thermodynamic stability of the MADE is enabled by the solvent-anchor complex, which exhibits excellent high-voltage stability and elevated levels of safety. This work presents a new perspective for developing high-voltage and high-safety Li metal batteries as well as gives significant insights into the role of molecular interactions on the electrochemical behavior of electrolytes. Further optimization of the MADE is underway to address critical challenges of the safe operation of practical Li metal batteries for extended service life under stringent conditions.

## Methods

### Materials

Single crystal LiNi$_{0.8}$Co$_{0.1}$Mn$_{0.1}$O$_2$ (sc-NMC811) cathode material was obtained from MTI. LiFSI (99%) was kindly provided by Nippon Shokubai (Japan). Li metal chips (450 μm) were purchased from China Energy Lithium Co. Ltd. 1,2-dimethoxyethane (DME, 99.9%) was obtained from Sigma. 2,2,3,3-Tetrafluoro-1-(1,1,2,2-tetrafluoroethoxy) propane (TTE, >98%), Dimethyl carbonate (DMC, 99%), 2-Methoxyethyl ether (G2, 99%), 1,1,2,2-Tetrafluoroethyl 2,2,2-Trifluoroethyl Ether (HFE, >97%) were obtained from MACKLIN. Carbon tetrachloride (CCl$_4$, 99.5%) was obtained from Sinopharm Group Co., Ltd. Deuterated dimethyl sulfoxide (DMSO-$d_6$, 99.9% D) and deuterated tetrahydrofuran (THF-$d_8$, 99.5% D) were obtained from Aladin. All solvents are dried with a 4 Å molecular sieve for more than 48 h. Lithium hexafluorophosphate (LiPF$_6$, 99.5%), bistrifluoromethanesulfonimidate (LiTFSI, 99%) were obtained from DADO New Material. Celgard 2500, Al foil, spring piece, gasket, and cell case (2032) were purchased from Canrd. The electrolytes were prepared by dissolving the lithium salt LiPF$_6$, LiFSI, or LiTFSI in the selected solvents inside a glovebox full of inert argon gas, where the water and oxygen content was <0.01 ppm. The NMC811 cathodes were prepared by mixing NMC811 active materials, super P Li (Timcal), and polyvinylidene fluoride (HSV900, MTI) in N-methyl-2-pyrrolidone (NMP, Aladdin) with a ratio of 8:1:1 to make a slurry before coating on Al foil. The cathodes were first dried at 60 °C for 4 h and then in a vacuum oven at 110 °C for 12 h before use. The NMC811 cathode average active material loading was 7.5 mg cm$^{-2}$.

### Electrochemical measurement

CR2032 coin cells were assembled for the electrochemical tests in an argon-filled glovebox, and 100 μL electrolyte was added inside before it was crimped. LSV, CA, and CV tests were carried out on BioLogic potentiostat, and the galvanostatic cycling tests were performed on LAND and NEWARE instruments. In the coin cells, the cathode (NMC811) loading is 7.5 mg cm$^{-2}$, and the anode (lithium metal) has a thickness of 450 μm. Celgard 2500 is used as the separator. For the pouch cell, the areal loading of the cathode is 2.5 mAh cm$^{-2}$, the anode employs a 50 μm lithium foil without a copper current collector and the N/P ratio is 2.1. The detailed pouch cell parameters are shown in Supplementary Table 1. The anodic stability was measured by LSV from initial potential to 5 V with a scan rate of 0.1 mV s$^{-1}$ using Li||PVDF + SP| Al cells. The cells were first cycled at 2.8–4.5 V three times at 0.1 C and

then charged to 4.5 V and kept at 4.6 and 4.7 V for 8 h, respectively, to test for leakage current. For CV tests, all cells were cycled between 2.8-4.7 V with a scan rate of 0.1 mV s$^{-1}$. For the Li||NMC811 cells, after the initial two activation cycles at C/10 (1 C = 200 mAh g$^{-1}$), the cells were subsequently cycled at C/3 between 2.8–4.7 V. The CE were measured in Li||Cu cells with a current density and areal capacity of 0.5 mA cm$^{-2}$ and 1 mAh cm$^{-2}$, respectively. Lithium deposition morphology was collected from Cu foil with a current density and areal capacity of 0.5 mA cm$^{-2}$ and 4 mAh cm$^{-2}$, respectively. All electrochemical tests were conducted at 25 °C.

### Characterizations

NMR was tested by Bruker AVAVCE III HD400. The deuterated reagent (DMSO-$d_6$) is encapsulated in a capillary tube as an external reference without direct interaction with the electrolyte. The residual solvent peak was used for peak referencing. The two-dimensional $^1$H-$^{19}$F nuclear magnetic resonance employs heteronuclear Overhauser effect spectroscopy (HOESY) mode. For FT-IR tests of H-bonding interactions between TTE with deuterated solvents, THF-$d_8$ or DMSO-$d_6$ was mixed with TTE in CCl$_4$ solution with ratios of 8:4.5:10 (THF-$d_8$: TTE: CCl$_4$) and 7:4.5:10 (DMSO-$d_6$: TTE: CCl$_4$), respectively. Raman mapping was conducted by LabRam HR Evolution. The ionic conductivity was tested using BioLogic VMP-3 and temperature variable oven (DHT test). The calculation of electrolyte ionic conductivity was through the resistance of 1 M KCl at 25 °C. Scanning electron microscopy (SEM, Gemini SEM 450) was used to observe the sample morphology. X-ray photoelectron spectroscopy (XPS) sputtering tests were performed on an ESCALAB 250Xi to characterize sample surface components. For HR-TEM (Talos F200X), the Al current collector of the cathode was peeled off, and the cathode sample was dispersed by ultrasonic in N, N-Dimethylacetamide. About the sample preparation of CEI and SEI, coin cells using different electrolytes were disassembled after 50 cycles, and the electrodes were extracted. Subsequently, the electrodes were rinsed with anhydrous DME five times (1 mL each time), to remove any residual electrolytes. Finally, the electrodes were subjected to vacuum drying. The thermal effects of mixing DME or HCE with TTE were tested by Isothermal Titration Calorimetry (VP-ICT). In situ XRD was performed with Cu Kα radiation (λ = 1.5418 Å) in the 2θ range 10°–50° using a Bruker diffractometer to characterize the evolution of sample structure. The mold was purchased from BJSCISTAR. After activating in MADE-1 and LHCE, the cathodes were recovered and tested using 1 M LiPF$_6$ in EC/DMC/EMC electrolyte to investigate the effect of surface CEI on the evolution of electrode structure. DSC was carried out by adding 30 mg of electrolyte and 2 mg Li metal or 4 mg 4.6 V-charged cathode material into tightly sealed high-pressure crucibles before being heated at a rate of 5 °C min$^{-1}$. The crucible material of DSC is gold-plated stainless steel, with a diameter of 6.35 mm, a thickness of 0.1 mm in the middle gasket, and a protrusion thickness of 0.5 mm. The ARC was tested by Extended Volume Adiabatic Acceleration Calorimeter (EV-ARC), and the pouch cell (Li||NMC811, 2 Ah, EVE Energy) with 3 g Ah$^{-1}$ electrolyte was first charged to the cut-off voltage of 4.6 V at 0.1 C to induce the formation of electrode-electrolyte interphases before the ARC test. An electric wire is fixed on the surface of the battery to measure the temperature changes of the battery. The temperature gradient is 5 °C, and the waiting time is 30 min. Soft X-ray absorption spectroscopy was performed at beamlines BL12B-a and BL12B-b of the NSRL in Hefei, China. The photon energy coverage was from 50 to 1000 eV, the resolving power (E/ΔE) is ≈2000, and the photon flux is 5 × 10$^9$ photons per second.

### MD simulation

The molecular dynamics simulations were conducted in the Forcite module in Materials Studio. All the dynamics simulations used the COMPASS II force field. For MADE-1, the box contains 11 LiFSI, 100 DME, and 300 TTE molecules. For MADE-3, the box contains 33

LiFSI, 100 DME, and 300 TTE molecules. For LHCE, the box contains 100 LiFSI, 120 DME, and 300 TTE molecules. The electrolyte systems were first simulated annealing between 600 and 300 K using an NTV ensemble for 15 cycles and optimized the structure after each annealing. The structure with the lowest energy is selected for the next dynamics simulations. The electrolyte systems were equilibrated in the isothermal-isobaric ensemble (constant NPT) using the Berendsen barostat to maintain the pressure of 0.0001 GPa for 500 ps, aiming to make the systems have proper density. Then, the last frame from the NPT simulation ran for 800 ps in the canonical ensemble (NVT) at 298 K. Thereafter, the radial distribution function was obtained by dynamic simulation of the optimized model and calculated the molecular coordination number by the following formula:

$$n(r') = 4\Pi\rho \int_0^{r'} g(r)r^2 dr \qquad (1)$$

$\rho$ is the Number density of certain types of atoms. The atomic coordinates of the initial and final configurations of the MD simulation could be found in the source data.

### Density-functional theory (DFT) calculation

The DFT calculation was conducted on the Gaussian16 package[61]. The initial geometry structures are derived from molecular dynamics simulations. All molecules and complexes were optimized by the B3LYP method at 6−311++G (d,p) basis set. Frequency analysis was performed to guarantee that all the structures were the minimum points on the potential energy surface. Calculation of oxidation potential were performed under an implicit solvation model with dielectric properties of Aniline (dielectric constant = 7.3). The oxidation potential $E_{ox}$ vs. Li/Li$^+$ was calculated as follows:

$$E_{OX} = \frac{G(M^+) - G(M)}{F} - 1.4V \qquad (2)$$

where G(M) and G(M$^+$) are the free energy of species M, its oxidized forms at 298.15 K, respectively. F is the Faraday constant. The deformation charge density was calculated using Dmol3 in Materials Studio software. The extreme points of ESP are achieved through Multiwfn software[45] combined with VMD programming. The atomic coordinates of the optimized geometries could be found in the source data.

### Reporting summary

Further information on research design is available in the Nature Portfolio Reporting Summary linked to this article.

## Data availability

The experiment data that support the findings of this study are available from the corresponding authors upon reasonable request. Source data are provided with this paper.

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

## Acknowledgements

This study was supported by the National Natural Science Foundation of China, grant Nos. 22179124 (X.R.), 21905265 (X.R.), CAS Project for Young Scientists in Basic Research, grant No. YSBR-098 (Q.W.), the Fundamental Research Funds for the Central Universities, grant No. WK3430000007 (X.R.), and 2021 Anhui Energy Internet Joint Fund Project, grant No. 2108085UD04 (Q.W. and X.R.). XAS tests were performed at beamlines BL12B-a and BL12B-b in the National Synchrotron Radiation Laboratory (NSRL) in Hefei, China. The numerical calculations in this paper have been done at Hefei Advanced Computing Center. In addition, the authors are grateful for resources from the High Performance Computing Center of Central South University and the Center for Micro and Nanoscale Research and Fabrication at USTC. We are also thankful for the helpful discussion about NMR tests and analysis with Dr. Ke Gong from the Instruments Center for Physical Science at USTC.

## Author contributions

Z.C. and X.R. conceived the idea and designed the experiments. Z.C. performed the main research work with the help of D.R., Q.N., J.F., and S.C. Z.J. and Q.W. conducted the ARC tests. Z.C., D.W., and X.O. performed the theoretical calculations. Z.H. and S.J. conducted the AFM test. J.J. performed the Raman test. J.M. carried out the DSC test. Z.C. and X.R. wrote the manuscript, and all authors discussed the results and revised the manuscript.

## Competing interests

The authors declare no competing interests.
