## [Peer Review File · Nature Communications]

Molecular Anchoring of Free Solvents for High-Voltage and High-Safety Lithium Metal BatteriesEditorial Note: Responses to Comment 5) of Reviewer #1 in the first-round of this Peer Review File have been redacted as indicated to maintain the confidentiality of unpublished data. Table R2 in the first-round responses to Reviewer #2 have been redacted as indicated to remove third-party material where no permission to publish could be obtained.

REVIEWER COMMENTS

Reviewer #1 (Remarks to the Author):

Cui et. al. proposed a new molecular anchoring approach to modulating the reaction activity of free ether solvents in dilute electrolytes. The formulated molecular anchoring dilute electrolyte (MADE) of LiFSI/DME/TTE (1:9:27) not only enables stable charge-discharge cycling of NCM811/Li battery up to 4.7V but also postpones the thermal runaway of the battery. The finding of this work is very interesting, which will promote the development of high-voltage and high-safety Li batteries. This work is suitable for the journal of Nature Communications after addressing the following issues.

- 1) Fig. 2c,f contained data of MADE 1, 2 and LHCE, while the subsequent studies mainly referred to MADE 1, 3 and LHCE. For the consistency of research content, I suggest to add the corresponding data of MADE 3 in Fig. 2c,f.
- 2) Please test the Aurbach CEs of the studied electrolytes, which is very simple and good to compare with the other reports.
- 3) From the result of average CE (Fig. 5a), MADE 1 is not the optimal one. The electrolyte design consideration on the anode side seems somehow different from the cathode side. Please give some discussion about that in the context.
- 4) TTE shows a strong H-bonding interaction with DME. Besides TTE, any other solvents have similar anchoring effects?
- 5) Can this kind of molecular anchoring dilute electrolytes enable stable lithium intercalation/de-intercalation of the graphite electrode?
- 6) Some figures need improvements. The unit of heat flow (uJ/s) should be revised to uJ/s/mol DME. The coordination number of 3 (Fig. 1f) is too small for a conventional dilute solution (DE). The information of Fig. 4g is not very clear.
- 7) Some information is missing. The electrolyte formula of LHCE and DE are not given. The length scale in Fig. 5d is not given. How to rinse (prepare) the samples of CEI and SEI? How many cycles operated for these CEI and SEI samples? Specify the parameters of 2 Ah pouch cell, such as the areal capacity and N/P ratio.

Reviewer #2 (Remarks to the Author):

“Molecular Anchoring of Free Solvents for High-Voltage and High Safety Lithium Metal Batteries” authored by Cui et al. attempts to describe a “molecular anchoring approach” to reduce the interfacial reactivity of free solvents in the dilute electrolyte “MADE”. This topic is highly interesting and worth investigating. However, the new concept of “MADE” and the molecular anchoring approach are based on the assumption that TTE forms hydrogen bond with DME and “reduce” free DME molecules, which has not been confirmed properly. More solid evidence must be provided since this H-bond is not conventional. To say the least, the evidence to confirm the existence of such H-bond was far from enough. I don't think the article can be accepted in its current form, the authors need to conduct more experiments to confirm the nature of “H-bond” between those two compounds.

1. First of all, the claim that hydrogen bond exists between DME and TTE is the core for the “MADE” concept. In an attempt to evaluate the hydrogen bond formation, the authors tried to use NMR spectroscopy, but they provided very few details about the NMR experiments, which significantly weakened the validity of the NMR data. In their experimental section, only two sentences “NMR was tested by AVANCE III HD400. The deuterated reagent (d₆-DMSO) is encapsulated in a capillary tube without direct interaction with the electrolyte” were provided without any further details. This is not acceptable if the authors really want to properly characterize the H-bond. Are all the peaks in Figure 1b referenced to DMSO-d₆? If there was no reference point, the chemical shifts cannot be compared. The authors should provide the DMSO-d₆ peak at 2.50 ppm for reference. It is well-known that there may be magnetic field drift in the NMR machine and thus, it is important for the authors to align (referenced) all the spectra to DMSO-d₆. Also, the authors claimed “When DME was mixed with TTE, the chemical shifts of H(TTE) increased while those of H(DME) decreased, indicating that H(TTE) acts as the H-bond donor with O(DME) as

the acceptor”, yet, in Figure 1b, the change in chemical shifts of DME protons is much larger than that of TTE protons! The result seems to contradict the claim that the protons of TTE are H-bond acceptors. Thus, DMSO-d₆ must be carefully labeled. Furthermore, the chemical shifts can be affected by other non-H-bond factors such as solvation and chemical environment. It is suggested that the authors also compare the chemical shifts of DME in solution such as DOL and DME. It is hard to imagine any H-bond can be formed between DOL and DME, yet, a large change in chemical shifts should be observed while mixing DOL with DME. Regarding the “two-dimensional 1H-19F nuclear magnetic resonance (NMR)” result, it is even harder to assess. Why did the authors use such a vague description? What was the sequence the authors used? HMQC? HSQC? HOESY? Scalar couplings? This experiment must be carefully presented to obtain valid data. The interpretation is somehow weird to say the least. The sentence “It was found that the pseudo H-bonds between C-H (DME)...F-C (TTE) is weaker than those between TTE molecules themselves” needs a lot of explanations. The protons of TTE are of course going to have very strong interaction with the F atoms, the F atoms are in fact splitting the proton signals as shown in the NMR spectra. Thus, the strong interaction between F atoms and protons of TTE is expected and there is nothing related to H-bond. Also, it seems that the 2D-NMR in Figure S2 has two different phases, what does it mean? Moreover, from Figure S2, it seems that DME protons are forming H-bond with the F atoms in TTE. Yet, a large upfield shift was observed for the DME protons. The results are not consistent with the formation of H-bond. The authors are trying to present a rather unconventional hydrogen bonding and very careful NMR characterization must be presented. Description such as “two-dimensional 1H-19F nuclear magnetic resonance (NMR)” is simply unacceptable. Indeed, there was evidence that the proton of CF₂H can be a H-bond acceptor (J. Am. Chem. Soc. 2017, 139, 9325–9332), but extensive characterization should be provided to validate the presence of this H-bond. It is suggested that the authors follow the characterization provided by 3 papers “J. Am. Chem. Soc. 2017, 139, 9325–9332”, “J. Am. Chem. Soc. 2000, 122, 4750-4755” and “J. Phys. Chem. B 2016, 120, 10679–10685” to properly characterize the existence of H-bond. Altogether, the validity of these NMR data is not high and should be properly processed.

2. Another piece of evidence for the claim of H-bond is the Raman spectroscopy. Normally, H-bond was characterized by the change of X-H stretching (J. Am. Chem. Soc. 2017, 139, 9325–9332) (in this case, C-H stretching). Why did the authors focus on this C-O-C symmetric stretching? What do the two peaks mean in pure DME? If there are two “free” DME C-O-C stretching in this region, why is there only one C-O-C stretching for Li⁺-DME? Assuming there is H-bond, when does the H⁺-DME behave the same as Li⁺-DME, given that Li⁺-DME bond is much stronger. The peaks must be properly assigned before we can draw any conclusion. Moreover, the sister IR spectra should be provided for comparison. The same changes in IR spectrum for C-O stretching should be expected. How about the C-H stretching region? Again, the formation of H-bond between TTE and DME as the “anchoring effect” is the major claim of this article and should be very carefully demonstrated experimentally.

3. There are also some minor points for the article. For example, the authors are using Al working electrode for LSV. It may work for LiPF₆ electrolyte due to passivation, but it is hard to ignore Al corrosion for LiFSI/LiTFSI electrolyte at high voltage. Another example is the sentence “Increasing the salt ratio in MADE-2 and MADE-3 results in improvements in Li CE, indicating that both TTE and FSI - anions are beneficial for Li metal protection”. I can understand why LiFSI is beneficial for the SEI formation. However, according to the authors, the TTE ratios among MADE-1, MADE-2 and MADE-3 didn't change, so, why TTE is also beneficial?

Reviewer #3 (Remarks to the Author):

In this work, the authors present a generic electrolyte design strategy using a molecular anchoring approach to suppress the reactivity of free solvents. The resulting molecular anchoring dilute electrolyte (MADE) is a low concentration electrolyte but retains high ionic conductivity. The electrolyte was well studied by experimental and computational methods, demonstrating the importance of the hydrogen bonding to the solvation structure and electrolyte stability. The Li||NCM811 coin cells and pouch cells were studied, demonstrating the high performance of MADE

for high-voltage Li-metal batteries. Moreover, MADE is also able to enhance the thermal stability of Li||NCM811 cells. I think the importance and novelty of this work are high, and I recommend it for publication after revision.

1. The authors claimed that the MADE fosters a more flexible SEI layer to accommodate volume changes of Li metal during cycling. Some discussion should be provided to explain this.
2. The MADE consists of a high TTE to DME molar ratio of 3:1. Why is the molar ratio of 3:1 used in this work?
3. More evidence should be provided to confirm the TTE-DME H-bond interactions. For example, the FTIR and ¹⁷O NMR can be employed to study the H bonding.
4. The testing conditions for Li||NMC-811 coin cells and pouch cells should be provided. For example, what is the ratio of anode to cathode? What is the thickness of Li metal anode?
5. The full electrochemical stability window of the MADE, showing Li plating and stripping, should be provided.
6. The Charge-discharge curves of Li||NMC-811 cells upon long-term cycling should be provided to show if there is voltage fade upon cycling.
7. How is the flammability of the MADE?
8. What is the reference peak for XPS?
9. The symmetric Li-Li cell test should be performed to assess the Li metal anode compatibility.
10. The Li cycling CE should also be evaluated using the Aurbach measurement.

Responses to reviewers' suggestions and comments

We would like to thank the reviewers for their valuable comments. We have incorporated most of the reviewers' comments and suggestions into the revised manuscript. We also provided detailed answers and explanations to the reviewers' other comments. The changes to the manuscript are marked **yellow** in this response and in the revised manuscript.

Reviewer #1:

Cui et. al. proposed a new molecular anchoring approach to modulating the reaction activity of free ether solvents in dilute electrolytes. The formulated molecular anchoring dilute electrolyte (MADE) of LiFSI/DME/TTE (1:9:27) not only enables stable charge-discharge cycling of NMC811/Li battery up to 4.7V but also postpones the thermal runaway of the battery. The finding of this work is very interesting, which will promote the development of high-voltage and high-safety Li batteries. This work is suitable for the journal of Nature Communications after addressing the following issues.

Response: We thank the reviewer for the positive comments on our work. We have incorporated additional experiments and analyses based on the reviewer's suggestions. Our point-to-point replies to the comments are detailed as follows.

1) Fig. 2c,f contained data of MADE 1, 2 and LHCE, while the subsequent studies mainly referred to MADE 1, 3 and LHCE. For the consistence of research content, I suggest to add the corresponding data of MADE 3 in Fig. 2c, f.

Response: We thank the reviewer for this kind suggestion. We have supplemented the cycling data, and leakage current data for MADE-3 (Fig. R1a-d). For cells using MADE-3, the capacity retention after 100 cycles is 86.2%, slightly lower than those for MADE-1 and MADE-2. This is because the side reaction of anion decomposition induced by DME at 4.7 V intensifies with the increase of the salt concentration, which would lead to the accumulation of by-products and increased interface resistance. Additionally, at 4.7 V, the leakage current of MADE-3 is slightly higher than MADE-1

and MADE-2 but lower than LHCE.

Figure R1. **a** Cycling performance of Li-NMC811 cells with MADE-1, MADE-2, MADE-3 and LHCE. **b** The leakage currents of Li||NMC811 cells at 4.6 V and 4.7 V. **c-d** Cycling performance and cell CE of three parallel tests for MADE-3.

We have added the above results to Figure 2 and the following discussion to the revised manuscript.

Page 11: "Specifically, the cell with MADE-2 exhibits excellent cycling stability, with a capacity retention of 95.0% over 100 cycles, whereas MADE-1 and MADE-3 retain 88.5% and 86.2% of the initial discharge capacity, respectively"

2) Please test the Aurbach CEs of the studied electrolytes, which is very simple and good to compare with the other reports.

Response: We appreciate the reviewer for this valuable suggestion. We followed reviewer's advice and compared the Aurbach Coulombic efficiency (CE) of the studied electrolytes. Preconditioning of the Cu electrode was performed by depositing and stripping of 2 mAh cm⁻² Li metal at 0.5 mA cm⁻². Then the Li CE was measured after 20 cycles of Li deposition/stripping (1 mAh cm⁻² each cycle, 2 mAh cm⁻² reservoir,

deposition at 0.5 mA cm^{-2} , stripping at 0.5 mA cm^{-2}).

Figure R2. The voltage curves of Li||Cu cells with Aurbach's method of DE (a), MADE-1 (b), MADE-2 (c), MADE-3 (d) and LHCE (e).

As shown in Fig. R2, the average Li CE in the DE (LiFSI: DME=1:9 by mole) is 97.60%, while that of MADE-1 increases to 99.26%. This indicates that anchoring free solvent is advantageous for enhancing the CE of the electrolyte. Further increasing the salt content improves the CE of the Li||Cu battery (from an average of 99.26% in

MADE-1 to 99.43% in MADE-3). This is because the decomposition of anions can form an interfacial layer enriched with inorganic species, which have low electronic conductivity, high interfacial energy, and thus can further inhibit electrolyte reduction and dendrite growth (Liu et al. *Angew. Chem. Int. Ed.* **2021**, *60*, 3661). With a much higher salt concentration (from MADE-3 to LHCE), there is little change in the CE of the Li||Cu cells (from 99.43% to 99.44%). The high salt content would aggravate the consumption of Li metal by reactions with anion, resulting in only a minor change in the Li CE compared to MADE-3.

The above results have been added to the revised manuscript in addition to the following description:

Page 17: “The Li CE results from Aurbach’s method exhibit a similar trend (Supplementary Fig. 32). In LHCE, the clustering of anions introduced by a high salt content aggravate the consumption of Li metal by reactions with anion, resulting in only a minor change in the Li CE compared to MADE-3.”

3) From the result of average CE (Fig. 5a), MADE 1 is not the optimal one. The electrolyte design consideration on the anode side seems somehow different from the cathode side. Please give some discussion about that in the context

Response: We thank the reviewer for this insightful question. It is true that MADE-1 is not the optimal electrolyte for the Li anode, in terms of the average CE and cycle life. However, MADE-1 shows the highest oxidation onset potential on the cathode side. The reason for such difference lies in the different behavior for the Li anode and the cathode during cycling.

For the cathode, the dominant species in MADE-1 is the DME-TTE complex, therefore the oxidation of DME is effectively suppressed due to molecular anchoring by TTE. Meanwhile, the low content of FSI⁻ reduces the anion-related H-transfer reactions (Ren et al. *Proc. Natl. Acad. Sci. U.S.A.* **2020**, *117*, 46, 28603-28613). However, a controllable degree of FSI⁻ decomposition may have a beneficial effect on the CEI layer,

based on the slightly improved capacity retention in MADE-2 compared to MADE-1. A very thin CEI layer with LiF, Li₂O, LiSO_x, LiNO_x and etc. (with contribution from FSI⁻ decomposition) would have favorable ionic conductivity and passivation ability to shield the reactive electrode surface from labile “free” solvent molecules generated from dynamic motions in the electrolyte. Nevertheless, as the concentration of FSI⁻ anion increases in LHCE, the H-transfer reaction between the ether solvent molecule and FSI⁻ anion becomes highly preferred under high voltages. Excessive solvent and FSI⁻ decompositions take place on the cathode surface, thus inducing the excessive accumulation of side products (Fig. 3f) and the cathode decay.

On the anode side, the chemical/electrochemical stability, ionic conductivity, flexibility and etc. of the SEI layer have critical effects on the Li metal stability. The increase of Li CEs with FSI⁻ contents in MADEs is likely due to the formation of a more ionic conductive SEI, which would improve the uniformity of Li deposition and stripping. Previous studies also indicate the beneficial effect of SEI species from FSI⁻ decompositions (Zhao et al. *Chem.* **2023**, 9, 3, 682-697). However, excessive FSI⁻ decomposition in the LHCE yields a highly rigid SEI layer compared to those in the MADEs, which has an adverse effect on the Li CE. Similar to the cathode side, a controllable degree of FSI⁻ participation in the SEI formation process is preferred. Therefore, the balance between inorganic and organic species would be of great importance for the electrolyte design for Li metal anodes.

This following discussion has been added in the revised manuscript.

Page 18: “The changes of Li CEs with the LiFSI content indicates that a controllable degree of FSI⁻ decomposition during the SEI formation process in MADEs is preferred, which could promote the formation of a more ionic conductive SEI to improve the uniformity of Li deposition and stripping. However, excessive FSI⁻ decomposition in the LHCE yields a highly rigid SEI layer, which has an adverse effect on the Li CE. Similarly, slight FSI⁻ decomposition on the cathode could help shield the reactive electrode surface from labile “free” solvent molecules generated from dynamic exchanges in the electrolyte, as supported by the improved

cathode stability in MADE-2.”.

4) TTE shows a strong H-bonding interaction with DME. Besides TTE, any other solvents have similar anchoring effects?

Response: We thank the reviewer for this good question. We have found that other hydrofluoroether molecules, for example, 1H,1H,5H-perfluoropentyl-1,1,2,2-tetrafluoroethylether (OTE) and ethyl 1,1,2,2-tetrafluoroethyl ether (ETE) which have the same -CF₂H moiety (molecular structures shown in Fig. R3), show similar anchoring effects with DME. In contrast, heptafluoro-1-methoxypropane (HM) without -CF₂H moiety cannot exhibit such strong anchoring effect.

Firstly, we mixed DME with OTE, ETE and HM (molar ratio=1: 1) and compared the changes of chemical shifts in the ¹H-NMR spectra. As shown in Fig. R4, the ¹H chemical shifts of -CF₂H of OTE and ETE are 5.4 and 5.5 ppm, respectively, while that of -CH₃ of HM is 3.5 ppm. After mixing with DME, the -CF₂H signals of OTE and ETE shift apparently to 5.8 ppm, while the -CH₃ signal of HM barely shifts to 3.7 ppm. This indicates that the anchoring effect between ETE/OTE and DME is stronger than that between HM and DME.

Secondly, we compared their differences in anodic stability by measuring the CV curves of Li||NMC811 cells in MADEs with different hydrofluoroethers. The molar ratios of salt, DME, and the anchoring agent are fixed at 1:9:27. As shown in Fig. R5, MADE-1(OTE) and MADE-1(ETE) exhibit no excessive oxidation side reactions during CV scans at 4.7 V, indicating that OTE and ETE with the -CF₂H moiety have similar anchoring effect. In contrast, increasing oxidation currents and higher leakage currents were observed with the use of HM, suggesting that the unique role of the -CF₂H moiety

Figure R3. The molecular structures of different hydrofluoroethers.

Figure R4. The ¹H-NMR spectra of different solvents and mixtures (*d*₆-DMSO as the external reference).

Figure R5. a-c CV curves of Li||NMC811 cells with different electrolytes at 0.1 mV/s. **d** The leakage currents of Li||NMC811 cells with different electrolytes at 4.6 V and 4.7 V.

We have added the above result and the following discussion to the revised manuscript.

Page 11: “Apart for TTE, we also found other hydrofluoroethers, such as 1H,1H,5H-perfluoropentyl-1,1,2,2-tetrafluoroethylene ether (OTE) and ethyl 1,1,2,2-tetrafluoroethyl ether (ETE) with the -CF₂H moiety, have similar anchoring effect (Supplementary Fig. 19–Fig. 20).”

Supporting information, Fig. 19: “The ¹H chemical shifts of -CF₂H of OTE and ETE are 5.4 and 5.5 ppm, respectively, while that of -CH₃ of HM is 3.5 ppm. After mixing with DME, the -CF₂H signals of OTE and ETE shift apparently to 5.8 ppm, while the -CH₃ signal of HM barely shifts to 3.7 ppm. This indicates that the anchoring effect between ETE/OTE and DME is stronger than that between HM and DME”

Supporting information, Fig. 20: “The anodic stability with different MADEs was estimated by CV. The molar ratios of salt, DME, and the anchoring agent are fixed at 1:9:27. MADE-1(O TE) and MADE-1(ET E) exhibit no excessive oxidation side

reactions during scans between 2.8-4.7 V, indicating that OTE and ETE with the $\text{-CF}_2\text{H}$ moiety have similar anchoring effect. In contrast, increased oxidation currents and higher leakage currents were observed with the use of HM, suggesting that the unique role of the $\text{-CF}_2\text{H}$ moiety.”

5) Can this kind of molecular anchoring dilute electrolytes enable stable lithium intercalation/de-intercalation of the graphite electrode?

[Redacted]

[Redacted]

[Redacted]

6) Some figures need improvements. The unit of heat flow (ucal/s) should be revised to ucal/s/mol DME. The coordination number of 3 (Fig. 1f) is too small for a conventional dilute solution (DE). The information of Fig. 4g is not very clear.

Response: We thank the reviewer for this good question. We have corrected this ucal/s to cal/s/mol DME and recalculated the heat (as shown in Fig. R8). This figure has been added to the revised manuscript. For high-concentration electrolytes, when DME is coordinated with lithium, the heat generated upon mixing with TTE significantly decreases. This indicates a competitive relationship between Li^+ and TTE for coordination with DME. Moreover, the coordination number of 3 in the schematic diagram (Fig. 1f) is appropriate because one DME molecule involves the coordination of two oxygen atoms with Li ions. In addition, we have revised Fig. 4g to make it clearer, as shown in Fig. R9.

Figure R8. Heat flow curves with DME and HCE added into TTE.

Figure R9. Schematic comparison of the cathode-electrolyte interfaces in LHCE and MADEs.

The above figures have been added to the revised manuscript.

7) Some information is missing. The electrolyte formula of LHCE and DE are not given. The length scale in Fig. 5d is not given. How to rinse (prepare) the samples of CEI and SEI? How many cycles operated for these CEI and SEI samples? Specify the parameters of 2 Ah pouch cell, such as the areal capacity and N/P ratio.

Response: We appreciate the reviewer's kind suggestion. The formulations of LHCE and DE electrolyte are LiFSI: DME: TTE=1:1.2:3, LiFSI:DME=1:9 (in molar ratio), respectively. Moreover, the length scale in Fig. 5d is 2 μm . We have re-emphasized this information in the revised manuscript.

Additionally, we have supplemented the description of the sample preparation procedures of CEI and SEI, the number of cycles and the pouch cell parameters (Table R1) in the supporting information of the revised manuscript.

“About the samples preparation of CEI and SEI, coin cells using different electrolytes were disassembled after 50 cycles, and the electrodes were extracted. Subsequently, the electrodes were rinsed with anhydrous DME five times (1 mL each time), to remove any residual electrolytes. Finally, the electrodes were subjected to vacuum drying. About the pouch cell, the areal loading of the cathode is 2.5 mAh cm^{-2} , the anode employs a $50 \text{ }\mu\text{m}$ lithium foil without a copper current collector. The N/P ratio is 2.1.”

Table R1. Cell parameters of the Li||NMC811 pouch cell.

	Parameter	Value
NMC811 cathode	Capacity	2.1 Ah
	Active material loading	96.5%
	Area capacity (each side)	2.5 mAh cm^{-2}
	Al foil thickness	$12 \text{ }\mu\text{m}$
Li anode	total thickness without Cu foil	$50 \text{ }\mu\text{m}$
N/P		2.1
Electrolyte	E/C	3 g Ah^{-1}
Separator	PE	$12 \text{ }\mu\text{m}$

Reviewer #2 (Remarks to the Author):

“Molecular Anchoring of Free Solvents for High-Voltage and High Safety Lithium Metal Batteries” authored by Cui et al. attempts to describe a “molecular anchoring approach” to reduce the interfacial reactivity of free solvents in the dilute electrolyte “MADE”. This topic is highly interesting and worth investigating. However, the new concept of “MADE” and the molecular anchoring approach are based on the assumption that TTE forms hydrogen bond with DME and “reduce” free DME molecules, which has not been confirmed properly. More solid evidence must be provided since this H-bond is not conventional. To say the least, the evidence to confirm the existence of such H-bond was far from enough. I don’t think the article can be accepted in its current form, the authors need to conduct more experiments to confirm the nature of “H-bond” between those two compounds.

Response: We greatly thank the reviewer for the insightful comments. Your suggestions and comments are very helpful for us. We also understand the reviewer’s concern about the H-bond interaction between DME and TTE, and we will provide a detailed response to your inquiries and make improvements to the manuscript based on your suggestions.

1. First of all, the claim that hydrogen bond exists between DME and TTE is the core for the “MADE” concept. In an attempt to evaluate the hydrogen bond formation, the authors tried to use NMR spectra, but they provided very few details about the NMR experiments, which significantly weakened the validity of the NMR data. In their experimental section, only two sentences “NMR was tested by AVAVCEIII HD400. The deuterated reagent (d6-DMSO) is encapsulated in a capillary tube without direct interaction with the electrolyte” were provided without any further details. This is not acceptable if the authors really want to properly characterize the H-bond. Are all the peaks in Figure 1b referenced to DMSO-d6? If there was no reference point, the chemical shifts cannot be compared. The authors should provide the DMSO-d6 peak at 2.50 ppm for reference. It is well-known that there may be magnetic field drift in the NMR machine and thus, it is important for the authors to align (referenced) all the

spectra to DMSO-d₆. Also, the authors claimed “When DME was mixed with TTE, the chemical shifts of H(TTE) increased while those of H(DME) decreased, indicating that H(TTE) acts as the H-bond donor with O(DME) as the acceptor”, yet, in Figure 1b, the change in chemical shifts of DME protons is much larger than that of TTE protons! The result seems to contradict the claim that the protons of TTE are H-bond acceptors. Thus, DMSO-d₆ must be carefully labeled. Furthermore, the chemical shifts can be affected by other non-H-bond factors such as solvation and chemical environment. It is suggested that the authors also compare the chemical shifts of DME in solution such as DOL and DME. It is hard to imagine any H-bond can be formed between DOL and DME, yet, a large change in chemical shifts should be observed while mixing DOL with DME. Regarding the “two-dimensional ¹H-¹⁹F nuclear magnetic resonance (NMR)” result, it is even harder to assess. Why did the authors use such a vague description? What was the sequence the authors used? HMQC? HSQC? HOESY? Scalar couplings? This experiment must be carefully presented to obtain valid data. The interpretation is somehow weird to say the least. The sentence “It was found that the pseudo H-bonds between C-H (DME)...F-C (TTE) is weaker than those between TTE molecules themselves” needs a lot of explanations. The protons of TTE are of course going to have very strong interaction with the F atoms, the F atoms are in fact splitting the proton signals as shown in the NMR spectra. Thus, the strong interaction between F atoms and protons of TTE is expected and there is nothing related to H-bond. Also, it seems that the 2D-NMR in Figure S2 has two different phases, what does it mean? Moreover, from Figure S2, it seems that DME protons are forming H-bond with the F atoms in TTE. Yet, a large upfield shift was observed for the DME protons. The results are not consistent with the formation of H-bond. The authors are trying to present a rather unconventional hydrogen bonding and very careful NMR characterization must be presented. Description such as “two-dimensional ¹H-¹⁹F nuclear magnetic resonance (NMR)” is simply unacceptable. Indeed, there was evidence that the proton of CF₂H can be a H-bond acceptor (J. Am. Chem. Soc. 2017, 139, 9325–9332), but extensive characterization should be provided to validate the presence of this H-bond.

It is suggested that the authors follow the characterization provided by 3 papers “J. Am. Chem. Soc. 2017, 139, 9325–9332” , “J. Am. Chem. Soc. 2000, 122, 4750-4755” and “J. Phys. Chem. B 2016, 120, 10679–10685” to properly characterize the existence of H-bond. Altogether, the validity of these NMR data is not high and should be properly processed.

Response: We thank the reviewer for the good questions and constructive comments. We will address the reviewer's questions point by point.

1. Details of the preparation of NMR Samples for tests

To avoid interference with the electrolyte, deuterated reagents (e.g., CDCl_3 and d_6 -DMSO) were sealed within the capillary inside the NMR tube under a torch as the external reference (as shown in Fig. R10).

Figure R10. The schematic diagram of NMR tests with external reference.

2. Peak referencing of deuterated reagents in NMR

We apologize for not showing the peak positions of deuterated reagents in Figure 1b of the manuscript and for mislabeling the deuterated reagent in the original supplementary information. In Fig. 1b of our original manuscript, deuterated chloroform ($d\text{-CDCl}_3$) was actually used as the external reference, and the characteristic peak of residual chloroform appears at 7.26 ppm (Fig. R11a). Additionally, we also supplemented the

NMR data using d_6 -DMSO as the external reference in Fig. R11b. Please note that the chemical shift difference of DME with the two external references is due to their differences in magnetic susceptibility. IUPAC recommends that the chemical shifts induced by such magnetic susceptibility effect be substrated when using external references (*Pure Appl. Chem.* 2008, 80, 59-84). Nevertheless, as shown in Fig. R11, the changes in chemical shifts for different solvents and electrolytes are basically consistent with different external references.

Figure R11. 1H NMR spectra of solvents and electrolytes with d - $CDCl_3$ (a) and d_6 -DMSO (b) as the external reference.

The above results and the experimental detail have been added to the revised manuscript.

3. Explanation of our statement that “When DME was mixed with TTE, the

chemical shifts of H(TTE) increased while those of H(DME) decreased, indicating that H(TTE) acts as the H-bond donor with O(DME) as the acceptor.”.

Please note that the hydrogen atom in TTE acts as the hydrogen bond donor, instead of TTE as the hydrogen bond acceptor. Upon the formation of hydrogen bonds, hydrogen atoms in hydrogen bond donors shown a higher chemical shift (decreasing in electron cloud density) due to the influence of hydrogen bond acceptors. This phenomenon is common and frequently reported in the literature (Sessler et al. *J. Am. Chem. Soc.* 2017, 139, 9325–9332; Shanahan et al. *J. Am. Chem. Soc.* 2020, 142, 19, 8809–8817; Arnold et al. *J. Am. Chem. Soc.* 2000, 122, 51, 12835–12841). In “Definition of the hydrogen bond” (IUPAC Recommendations 2011, *Pure Appl. Chem.* 2011, 83, 1637-1641), it also describes that “the X–H...Y–Z hydrogen bond leads to characteristic NMR signatures that typically include pronounced proton deshielding for H in X–H, through hydrogen bond spin–spin couplings between X and Y, and nuclear overhauser enhancements.” In our ¹H NMR results, the H atom of TTE exhibits an increased chemical shift (deshielding), in consistent with the hydrogen bond donor behavior. However, if the H atoms of DME form dominant hydrogen bonds with the F atoms of TTE, the corresponding H atom of DME should also experience an increased chemical shift, which is not in line with our experimental result.

On the other hand, a certain degree of correlation between H_(DME) and F_(TTE) was indeed observed in the ¹H-¹⁹F HOESY NMR spectrum. We anticipate that there would be two factors influencing the chemical shift of hydrogen atoms in DME simultaneously. Firstly, the H-bond anchoring interactions between H_(TTE) and O_(DME) bring TTE and DME to a close proximity and significantly change the charge distribution of DME (similar to the coordination of Li⁺ with DME in LHCE), which is the main reason for the ¹H chemical shift of DME to a lower value. In the meantime, the spatial proximity of H_(DME) and F_(TTE) induces relatively weak interactions, which would potentially cause the ¹H chemical shift of DME to a higher value (H_(DME) as the H-bond donor instead). Overall, the dominant force is the H-bond between H_(TTE) and O_(DME) and the observed

¹H chemical shift of DME is toward a lower value.

We have supplemented the following description in the revised manuscript.

Page 6: “Although the spatial proximity of H_(DME) and F_(TTE) induces relatively weak interactions, which would potentially cause the ¹H chemical shift of DME to a higher value (H_(DME) as the H-bond donor instead), the dominant force is the H-bond between H_(TTE) and O_(DME) and the resulting ¹H chemical shift of DME is toward a lower value.”

In the following parts, we will provide more comprehensive evidence to prove the presence of hydrogen bonds between DME and TTE.

4. Explanation of the question that change in chemical shifts of DME protons is much larger than that of TTE protons.

As shown in Fig. R11, when using *d*₆-DMSO as the deuterated reagent, the change of ¹H chemical shifts of DME is -0.379 ppm, while the ¹H chemical shift change in the -CF₂H of TTE is 0.233 ppm. It is worth noting that the ratio of TTE to DME is 3:1 here. And the NMR results represent the averaged outcomes of molecules in different states. In fact, a significant portion of TTE remains uncoordinated with DME, while all DME molecules are basically in the coordinated state. We further adjusted the molar ratio of DME to TTE from 1:1 to 1:5, as shown in Fig. R12a-b. When the ratio of DME to TTE is 1:1, the change in ¹H chemical shift of DME is -0.29 ppm, which is less than the change in the ¹H chemical shift of -CF₂H of 0.42 ppm. As the proportion of TTE gradually increases, the ¹H chemical shift of DME continues to shift toward lower chemical shifts, whereas the ¹H chemical shift of TTE moves toward pure TTE.

Figure R12. a ¹H NMR spectra of DME and TTE mixtures at different ratios (*d*₆-DMSO as the external reference). **b** Changes in ¹H NMR chemical shifts with varying ratios of DME to TTE.

5. Explain and compare the chemical shifts of DME in solution such as DOL and DME

We mixed DME and DOL in different molar ratios of 1:0.5, 1:1, 1:2, and 1:3 and conducted NMR tests (Fig. R13a). With the addition of DOL, the protons of DME shift toward a higher field (decrease in chemical shift). The protons of DOL shift toward a lower field, and as the content of DOL increases, the change in the chemical shift of DOL protons decreases. This indicates an interaction occurring between the protons (H-bond donor) in DOL and the oxygen atom in DME. In addition, weak and improper hydrogen bonds have been reported between C-H and C-O in ether (Tatamitani et al. *J. Am. Chem. Soc.* 2002, 124, 11, 2739-2743). We believe that this interaction exists

but is too weak to be termed as a conventional hydrogen bond. However, $-\text{CF}_2\text{H}$ as a hydrogen bond donor has been extensively studied. (Erickson et al. *J. Org. Chem.* 1995, 60, 6, 1626–1631; Zafrani et al. *J. Med. Chem.* 2019, 62, 11, 5628–5637; Malquin et al. *Chem. Commun.*, 2019, 55, 12487-12490; Shanahan et al. *J. Am. Chem. Soc.* 2020, 142, 19, 8809–8817; Zafrani et al. *J. Med. Chem.* 2017, 60, 2, 797–804; Box et al. *Chem. Sci.*, 2021, 12, 10252-10258). Compared to the hydrogen in the $-\text{CH}_2-$ moiety of DOL, the hydrogen in the CF_2H structure is less shielded, making it exhibit a stronger "acidity". Therefore, after coordinating with TTE, the hydrogen in DME experiences a greater chemical shift than coordinating with DOL (Fig. R13b).

We further measured the heat release during the mixing of DME and DOL using ITC. It was found that the heat flow is much lower than that between DME and TTE, indicating that stronger interactions exist between DME and TTE (Fig. R14).

Figure R13. **a** ^1H NMR spectra of DME and DOL mixtures at different ratios (d_6 -DMSO as the external reference). **b** Comparison of change in ^1H chemical shifts of DME after TTE coordination and DOL coordination.

Figure R14. a Heat flow curves with DME added into DOL. **b** Heat flow curves with DME and HCE added into TTE.

6. About the two-dimensional ^1H - ^{19}F tests and result interpretation.

We apologize for not clearly describing the experimental details of the two-dimensional ^1H - ^{19}F nuclear magnetic resonance (NMR). NMR tests were done with Bruker Avance III HD400. The deuterated reagent (*d*- CDCl_3) was encapsulated in a capillary tube to avoid direct interaction with the electrolyte. The two-dimensional ^1H - ^{19}F nuclear magnetic resonance employs Heteronuclear Overhauser Effect Spectroscopy (HOESY) sequence.

The reason for the two-phase ^1H - ^{19}F NMR spectra shown in our original manuscript is due to the use of phase sensitivity mode HOESY for the test. Similar behavior has also been reported in the literature (Liu et al. *Org. Chem. Front.* 2014, 1, 494-500). To avoid confusion to readers and for easier comparison of the strength of interaction, we have corrected the data using the absolute values for the 2D ^1H - ^{19}F NMR spectrum, as shown in Fig. R15.

Figure R15. Two-dimensional ^1H - ^{19}F NMR spectra of DME-TTE mixture (with CDCl_3 as the external reference).

As shown in the inset of Fig. R15, TTE exhibits four distinct chemical environments for fluorine atoms (F_1 , F_2 , F_3 , F_4) and three distinct chemical environments for hydrogen atoms. Among hydrogens of TTE, two H_1 atoms belong to $-\text{CF}_2\text{H}$ with similar NMR shifts. DME has two distinct groups of hydrogen atoms labeled as H_3 and H_4 . The ^1H - ^{19}F coupling signals were observed between H_1 and $\text{F}_1/\text{F}_2/\text{F}_3/\text{F}_4$, as well as H_2 and $\text{F}_1/\text{F}_2/\text{F}_3/\text{F}_4$. However, all of those signals are possibly from the short as well as long-range coupling between H and F atoms in the same TTE molecule. The coupling between H and F atoms between different TTE molecules cannot be easily distinguished. Although the interactions between C-H (DME)...F-C (TTE) could be observed (coupling signals between $\text{H}_{3/4}$ and $\text{F}_1/\text{F}_2/\text{F}_3/\text{F}_4$), they are not the dominant force between DME and TTE.

As we discussed earlier, although the spatial proximity of $\text{H}_{(\text{DME})}$ and $\text{F}_{(\text{TTE})}$ induces relatively weak interactions, which would potentially cause the ^1H chemical shift of DME to a higher value ($\text{H}_{(\text{DME})}$ as the H-bond donor instead), the dominant force is the H-bond between $\text{H}_{(\text{TTE})}$ and $\text{O}_{(\text{DME})}$ and the observed ^1H chemical shift of DME is

toward a lower value.

We have revised the description in the manuscript as follows.

Page 6: “Meanwhile, the dominant interaction between DME and TTE in MADEs is not from the pseudo hydrogen bond between H_(DME) and F_(TTE). Despite the ¹H-¹⁹F coupling signals observed in two-dimensional nuclear magnetic resonance (NMR), as shown in Supplementary Fig. 2, the pseudo H-bonds between C-H (DME)...F-C (TTE) are weaker than those between H_(TTE) and O_(DME).”

“Although the spatial proximity of H_(DME) and F_(TTE) induces relatively weak interactions, which would potentially cause the ¹H chemical shift of DME to a higher value (H_(DME) as the H-bond donor instead), the dominant force is the H-bond between H_(TTE) and O_(DME) and the observed ¹H chemical shift of DME is toward a lower value. Further geometry optimizations of DME-TTE complexes by density-functional theory (DFT) also suggest that the H_(TTE)...O_(DME) interactions are favored compared to H_(DME)...F_(TTE) interactions (Supplementary Fig. 3), which is consistent with previous studies (Zhang et al. *Molecular Physics*. 2014, 112, 13, 1736-1744)”

To prove that the major interaction site is not between H_(DME) and F_(TTE), we also present the following argument.

6.1 Previous literature

According to Zhang et al., the natural bond orbital (NBO) method is very useful for making bonding analyses of noncovalent interactions when donor-acceptor interactions take place (Zhang et al. *Molecular Physics*. 2014, 112, 13, 1736-1744). The stabilization energy caused by donor-acceptor interactions is generally estimated by the second-order perturbation energy $\Delta E_{i-j^*}^{(2)}$. The i and j^* indicate a lone pair orbital and an antibonding σ^* orbital, respectively. Their results indicated that the stabilization energy between C-H...F is significantly weaker than that of C-H...O when different ethers (C₂H₄O: oxirane, C₄H₈O₂: dioxane) interact with CHF₃ (Table R2).

Table R2 The second-order perturbation stabilization energies $\Delta E_{i-j}^{(2)}$ of $C_2H_4O-CHF_3$, $C_4H_6O-CHF_3$, $C_4H_8O_2-CHF_3$, and $C_5H_5N-CHF_3$. The calculations are performed at the B97D/6-311++G** (Copied from Zhang et al. *Molecular Physics*, 2014, 112, 13, 1736-1744).

[Redacted]

6.2 Miscibility tests for $-CF_2H$ and $-CF_3$ moieties

If the $H_{(DME)}$ have a strong interaction with F atoms, the DME would likely exhibit good miscibility with 1H-perfluorohexane, methylnonafluorobutyl ether and 2,2,3,3,3-pentafluoropropyl methyl ether in electrolyte. However, 1 M LiFSI-DME does not mix well with these molecules (Wu et al. *Chem*, 2023, 9, 3, 650-664). In particular, it should be noted that 2,2,3,3,3-pentafluoropropyl methyl ether (HFE-2) featuring a $-CF_3$ end group is immiscible with 1 M LiFSI-DME, while ETE with a $-CF_2H$ end group is fully miscible (Fig. R16). This result indicates that the C-H (ETE)...O-C (DME) interaction determines the miscibility between ETE and DME, while the C-H (DME)...F-C (hydrofluoroether) interaction is much weaker. This supports that the major interaction sites between HFE and DME are not between fluorine and hydrogen.

Figure R16. Miscibility of 1 M LiFSI in DME with 2,2,3,3,3-pentafluoropropyl methyl ether and ethyl 1,1,2,2-tetrafluoroethyl ether.

6.3 Theoretical calculations

Another piece of evidence is from DFT calculations (B3LYP, 6-311G++ (d, p)) (Fig. R17). We brought the $H_{(DME)}$ close to the $F_{(TTE)}$ on purpose and then conducted geometry optimizations by DFT. The optimization results showed that the $H_{(DME)}$ moved away from the $F_{(TTE)}$, while the $H_{(TTE)}$ moved closer to the $O_{(DME)}$. To further verify this result, we selected another structure in which the $H_{(DME)}$ and the $F_{(TTE)}$ were brought close, and both calculations exhibited similar outcomes. The distances between $H_{(DME)}$ and $F_{(TTE)}$ before optimization were 2.45 Å (2.44 Å in structure 2) and 2.57 Å (2.18 Å in structure 2), respectively. After optimization, the distances increased to 3.13 Å (5.52 Å in structure 2) and 3.43 Å (2.96 Å in structure 2). On the other hand, the distance between $O_{(DME)}$ and $H_{(TTE)}$ decreased from the original 4.11 Å (5.45 Å in structure 2) to 2.29 Å (2.35 Å in structure 2). These results also suggest that the C-H (TTE)...O-C (DME) interactions are favored compared to C-H (DME)...F-C (TTE) interactions.

Figure R17. DFT optimization of different configurations of the DME-TTE complexes.

7. The existence of hydrogen bonds between CF₂H and C-O-C

Although previous studies have already demonstrated the presence of a hydrogen bond between CF₂H and O in methoxymethane (CH₃OCH₃) (Delanoye, et al. *J. Am. Chem. Soc.* **2002**, 124, 40, 11854–11855; Nagels, et al. *Chemistry – A European Journal.* **2014**, 20, 27, 8433-8443; Mukhopadhyay, et al. *The Journal of Physical Chemistry A.* **2010**, 114, 14, 5026-5033;), here we provide more evidence of the hydrogen bond interaction between the CF₂H in TTE and C-O-C in DME.

7.1 Prove that CF₂H as hydrogen bond donor using FT-IR.

For an accurate determination of the presence of red-shift or blue-shift hydrogen bonds, it is necessary to prepare isolated individual molecules as well as complexed molecules for the investigation of infrared or Raman shifts. Here we dissolved TTE into CCl₄ (99.5%) to minimize intermolecular interactions in neat liquids (Sessler et al. *J. Am. Chem. Soc.* **2017**, 139, 27, 9325–9332). CCl₄ serves as a diluent to disrupt the original interactions between TTE molecules, allowing for the study of the hydrogen bonding of -CF₂H.

It is worth noting that due to the overlap between the C-H vibration signals of DME and TTE, deuterated DME (d_{10} -DME) is preferred to study the shift of $-\text{CF}_2\text{H}$ in TTE. However, due to export control policy, we are unable to obtain d_{10} -DME, so we are using d_8 -THF with very similar etheral moiety as a substitute.

Figure R18. Comparison of infrared spectra before and after mixing 2 M TTE with 5 M d_8 -THF and 5 M d_6 -DMSO in CCl_4 solution.

We chose 2 M TTE in CCl_4 as the control sample to study the shift of $\text{CF}_2\text{-H}$ bond vibration. As shown in Fig. R18, after adding d_8 -THF to the 2 M TTE, the $\text{CF}_2\text{-H}$ vibration of TTE shifts noticeably from 3000 to 3010 cm^{-1} , which is consistent with the characteristics of the blue-shift $-\text{CF}_2\text{H}\cdots\text{O}$ hydrogen bond observed by Sessler et al. (Sessler et al. *J. Am. Chem. Soc.* 2017, 139, 27, 9325–9332). This supports the evidence that the interaction site between deuterated THF and TTE is the $\text{H}_{(\text{TTE})}$ and the $\text{O}_{(\text{THF})}$.

In addition, we study the interaction between TTE and DMSO, which is often selected to study the $\text{C-H}\cdots\text{O}$ hydrogen bond as it contains no hydrogen donor (Salamone et al. *J. Org. Chem.* 2012, 77, 23, 10479–10487; Wang et al. *J. Phys. Chem. A.* 2003, 107, 23, 4683–4687; Box et al. *Chem. Sci.*, 2021, 12, 10252–10258). As shown in Fig. R18, upon adding d_6 -DMSO, a noticeable blue-shift of the $\text{CF}_2\text{-H}$ signal in TTE was also observed, which can demonstrate the hydrogen bond donating ability of TTE.

We have revised the description in the manuscript as follows.

Page 7:” To further verify the hydrogen bonding between TTE and ether molecules, we carried out Fourier-transform infrared spectroscopy (FT-IR) analysis in CCl₄ solutions. To study the effect of hydrogen bonding on the C-H vibration in TTE while avoiding the interference of C-H bonds in ether (overlapping signals), deuterated tetrahydrofuran (*d*₈-THF) was selected because of its characteristic ethereal moiety and easy accessibility. As shown in Supplementary Fig. 4, after adding *d*₈-THF to the TTE solution, the CF₂-H vibration peak shifts noticeably from 3000 to 3010 cm⁻¹, which aligns with the featured blue-shifting vibration of CF₂-H after H-bond formation reported in the literature^{38, 40, 41}. In addition, we study the interaction between TTE and DMSO, which is often selected to study the C-H...O hydrogen bond as it contains no hydrogen donor. Similarly, upon adding *d*₆-DMSO, a noticeable blue-shift of the CF₂-H signal in TTE was also observed, which can demonstrate the hydrogen bond donating ability of TTE.”

7.2 DFT calculations

Previously, Sessler et al. (*J. Am. Chem. Soc.* 2017, 139, 27, 9325–9332) studied the relationship between CF₂H...O bond lengths and bond angles in various solid crystals, and found a potential structural correlation between CF₂H and oxygen ($y = -6.5 \times 10^{-3}x + 3.15$, where y is the bond length, x is the bond angle). This correlation is useful for identifying the presence of hydrogen bond interaction between CF₂H and oxygen. We also used the MP2 method (6-311G++ (d, p)) to optimize the geometry of the DME-TTE complex (Fig. R19). We found that the bond length (2.17 Å) and bond angle (150.0°) for C-H (TTE)...O-C (DME) of the optimized DME-TTE complex match well with the above correlation equation.

Figure R19. The optimized DME-TTE complex.

7.2 ^1H - ^{13}C heteronuclear singular quantum correlation (HSQC) spectra

We attempted to confirm the hydrogen bonding interaction between CF_2H and O using the ^1H - ^{13}C heteronuclear singular quantum correlation (HSQC) spectrum (Chattopadhyay et al. *J. Phys. Chem. B.* 2016, 120, 41, 10679–10685). The methyl carbon and methylene carbon of DME are labeled as C1 and C3, respectively, while the methylene carbon of TTE and the carbon in CF_2H are labeled as C2 and C4, respectively. However, due to the low abundance of ^{13}C isotopes, only the correlation between carbon and hydrogen directly connected to carbon was observed in the two-dimensional spectrum, and no other enhanced signals were observed (Fig. R20).

Figure R20. Two-dimensional ^1H - ^{13}C HSQC NMR spectrum for the mixture with the ratio of DME:TTE=1:3 (d_6 -DMSO as external reference).

7.3 ^{17}O NMR

We further compared the ^{17}O NMR spectra of mixed solutions with different ratios of DME and TTE (Fig. R21). The results showed that as the content of TTE increased, the ^{17}O signal of DME shifted to a lower chemical shift, which agrees with the previous study that hydrogen bonding induces an upfield shift of ^{17}O -NMR (Reuben et al. *J. Am. Chem. Soc.* 1969, 91, 21, 5725). Meanwhile, there is no apparent shift of ^{17}O -NMR signal for the oxygen atom in TTE. Therefore, the hydrogen bonding between $-\text{CF}_2\text{H}$ and $\text{O}_{(\text{DME})}$ is further validated.

Figure R21 ^{17}O NMR spectra of mixed solutions with different ratios of DME and TTE.

7.4 Other evidence for the hydrogen bond between CF_2H and C-O-C

Here, in order to more fully demonstrate the existence of hydrogen bonds between DME and TTE, we are further adopting an alternative method of proof again. It is known that DME can form hydrogen bonds with aniline ($\text{H}_{\text{aniline}} \dots \text{O}_{\text{DME}}$), and TTE can form hydrogen bonds with d_6 -DMSO ($\text{H}_{\text{TTE}} \dots \text{O}_{d_6\text{-DMSO}}$) (Salamone et al. *J. Org. Chem.* 2012, 77, 23, 10479–10487; Wang et al. *J. Phys. Chem. A.* 2003, 107, 23, 4683–4687; Box et al. *Chem. Sci.*, 2021, 12, 10252–10258). If it can be confirmed that the impact of TTE on DME is consistent with the impact of aniline on DME, and the effect of DME on TTE aligns with the effect of d_6 -DMSO on TTE, then it can support that the hydrogen bond is situated between the H_{TTE} and the O_{DME} .

7.4.1 Comparison of the interactions of TTE and aniline with DME

C-H vibration

Pure DME has four peaks in the range of 2800-3000 cm^{-1} , located at 2981, 2927, 2878 and 2818 cm^{-1} , respectively. When TTE is added to the DME (note that neat liquids were used for mixing), the C-H vibration peak of the DME significantly shifts towards higher wave numbers (Fig. R22a). When replacing TTE with aniline and adjusting the ratio of DME to aniline (Fig. R22b), it can be observed that with increasing aniline content, the C-H vibrational peak of DME also shifts to a higher wavenumber, consistent with the impact of TTE on DME.

Figure R22. Comparison of the changes in the C-H vibrations of DME after DME mixing with aniline or TTE.

The ^1H NMR Chemical Shifts

Fig. R23 depicts the ^1H NMR spectra of DME mixed with aniline at a ratio of 1:3. Both TTE and aniline have a similar effect on the hydrogen atoms of DME. With adding TTE

or aniline to DME, the hydrogen atoms of DME shift towards lower chemical shifts.

Figure R23. The ^1H NMR spectra of DME, aniline and the mixture of DME: aniline=1:3 (d_6 -DMSO as the external reference).

7.4.2 Comparison of the interactions of DME and DMSO with TTE using ^1H NMR

As shown in Fig. R24, DME and d_6 -DMSO have a similar impact on TTE, causing a high chemical shift in TTE's hydrogen signals.

Figure R24. The influence of d_6 -DMSO on the NMR hydrogen spectrum of TTE (d_6 -DMSO as the external reference).

2) Another piece of evidence for the claim of H-bond is the Raman spectroscopy. Normally, H-bond was characterized by the change of X-H stretching (J. Am. Chem. Soc. 2017, 139, 9325–9332) (in this case, C-H stretching). Why did the authors focus on this C-O-C symmetric stretching? What do the two peaks mean in pure DME? If there are two “free” DME C-O-C stretching in this region, why is there only one C-O-C stretching for Li⁺-DME? Assuming there is H-bond, when does the H⁺-DME behave the same as Li⁺-DME, given that Li⁺-DME bond is much stronger. The peaks must be properly assigned before we can draw any conclusion. Moreover, the sister IR spectra should be provided for comparison. The same changes in IR spectrum for C-O stretching should be expected. How about the C-H stretching region? Again, the formation of H-bond between TTE and DME as the “anchoring effect” is the major claim of this article and should be very carefully demonstrated experimentally

Response: We thank the reviewer for the insightful comments and constructive suggestions. The main reason of focusing on the vibration peak of -CH₂-O-CH₃ is due to the overlap of C-H vibration signals in DME and TTE in our previous attempts to verify the H-bond. The study of -CH₂-O-CH₃ vibration also help elucidate the Li solvation structure in different electrolytes. The peaks at 821 cm⁻¹ and 848 cm⁻¹ in the Raman spectrum of pure DME correspond to the CH₂ rocking and C–O stretching vibrations, respectively (Xiao et al. *J. Am. Chem. Soc.* 2017, 139, 28, 9475–9478; Yoshida et al. *J. Phys. Chem. A* 1998, 102, 16, 2691–2699). From MADE-1 to LHCE, DME gradually transitions from being coordinated by TTE to being coordinated by Li⁺. There are two new bands at 839 and 874 cm⁻¹ after Li⁺ coordination (Cao et al. *Adv. Mater.* 2021, 33, 2103178). However, the peak at 839 cm⁻¹ overlaps with the peak of TTE at 837 cm⁻¹, making it difficult to distinguish. Therefore, we can only confirm that the vibration peak at 874 cm⁻¹ gradually strengthens, which indicates that the proportion of Li⁺-coordinated DME increases.

Additionally, C-O-C vibration peaks in the infrared spectra have been supplemented (Fig. R25). The characteristic C-O stretching band of pure DME is located at 1106 cm⁻¹

¹. The introduction of LiFSI into DME-TTE mixture produces a new peak at 1082 cm^{-1} , which gradually increases with the salt content, thus attributed to the coordination between C-O-C moieties and Li^+ . (Chen et al. *Angew. Chem.* 2022, 134, e202207645; Tian et al. *Adv. Sci.* 2022, 9, 2201207).

For changes of C-H vibration signals, please refer to Fig. R18 for the interaction between TTE and d_8 -THF in CCl_4 . As discussed earlier, the addition of d_8 -THF to TTE causes the featured blue-shift of the C-H bond in $-\text{CF}_2\text{H}$ with the formation of the H-bond between $\text{H}_{(\text{TTE})}$ and $\text{O}_{(\text{THF})}$.

Figure R25. The changes in the C-O-C stretching vibration of DME as a function of salt concentration.

We have revised the discussion in the manuscript as follows:

Page 8: “The two Raman peaks of pure DME at 821 cm^{-1} and 848 cm^{-1} are C-O stretching and CH_2 rocking signals^{48,49}. The vibration signal of free FSI⁻ anions (720 cm^{-1}) in the dilute electrolyte (DE, LiFSI-9DME in molar ratio) could easily be identified, while the anions exist in the form of contact ion pairs (CIP, 730 cm^{-1}) in MADEs. This is because the coordination of

TTE weakens the binding affinity between DME and Li⁺. As the salt concentration increases, the signals associated with Li⁺-coordinated DME (877 cm⁻¹, C-O stretching) and ion aggregates (AGG, 753 cm⁻¹, S-N-S stretching) gradually intensify^{50,51}. Similar changes in the IR spectrum are observed as the new peak associated with Li⁺-coordinating DME at 1079 cm⁻¹ becomes stronger with the increasing salt content⁵² (Supplementary Fig. 8), which agrees with our argument that the dominant species gradually transform from DME-TTE complexes in MADEs into Li⁺-DME complexes in LHCE.”

3) There are also some minor points for the article. For example, the authors are using Al working electrode for LSV. It may work for LiPF₆ electrolyte due to passivation, but it is hard to ignore Al corrosion for LiFSI/LiTFSI electrolyte at high voltage.

Another example is the sentence “Increasing the salt ratio in MADE-2 and MADE-3 results in improvements in Li CE, indicating that both TTE and FSI - anions are beneficial for Li metal protection”. I can understand why LiFSI is beneficial for the SEI formation. However, according to the authors, the TTE ratios among MADE-1, MADE-2 and MADE-3 didn’t change, so, why TTE is also beneficial?

Response: We appreciate the reviewer for the comments. We measured the corrosion current on Al foil using different salts with the DE, LHCE, and MADE solvations structure through CV scans. As shown in Fig. R26, diluent electrolytes (1 M LiFSI-DME and 1 M LiTFSI-DME) exhibit obvious corrosion of Al foil, and the corrosion currents tend to increase with the number of CV scans. In contrast, MADEs and LHCE electrolytes show significantly reduced corrosion currents, which may be attributed to the passivation effect of TTE on Al (Ren et al. *Chem.* 2018, 4, 8, 1877-1892). SEM morphology analysis of Al foils after three CV scans also reveals that Al foils tested with MADEs and LHCE are smoother compared to those tested with diluent electrolytes.

Figure R26. The corrosion behavior of aluminum in different electrolytes (scan rate: 0.1 mV/s).

Overall, the Al corrosion current is low (less than $2 \mu\text{A cm}^{-2}$) in MADEs and LHCEs, and this does not affect the determination of the onset potential in LSV. This result has been added in the revised manuscript.

Page 11: “Moreover, the Al corrosion current remains below $2 \mu\text{A cm}^{-2}$ in MADEs and LHCEs utilizing LiFSI or LiTFSI due to the passivation effect of TTE (Supplementary Fig. 13). No apparent Al corrosion could be observed from SEM images after CV scans.”

Explanation of why TTE is also beneficial for Li anode

The improvement of Li CE in the Li||Cu cells from DE to MADE-1 is due to the addition of TTE. Compared to the DE, the anchoring of DME by TTE would weaken the ability

of DME to coordinate with Li^+ , thus increasing the coordination between Li^+ and FSI $^-$. For example, in MADE-1, although its actual salt concentration is only 0.19 M, the anions still exist in the form of CIP (contact ion pair), rather than free anions.

Additionally, TTE itself can participate in the formation of the SEI, creating an interface layer rich in fluorinated species, which is beneficial for efficient Li deposition and stripping (Zheng et al. *Chem*, 2021. 7(9), 2312-2346. Kwak et al. *Adv. Funct. Mater.* 2021, 31, 2002927). We further studied the impact of TTE content on the Li CE under a fixed ratio of LiFSI: 9DME (Fig. R27). The results indicate that adding TTE can apparently improve the Li CE during deposition and stripping.

We have revised the description in the manuscript as follows.

Page 18 “TTE functions as the anchoring agent for DME molecules, weakening the ability of DME to coordinate Li^+ and amplifying the proportion of FSI $^-$ in the solvation sheath. Moreover, TTE participates in the formation of the SEI, resulting in an interface layer enriched with fluorinated species, which is beneficial for Li metal protection.”

Figure R27. The relationship between the Li CE and the TTE content during repeated tests (with LiFSI:DME ratio of 1:9).

Reviewer #3 (Remarks to the Author):

In this work, the authors present a generic electrolyte design strategy using a molecular anchoring approach to suppress the reactivity of free solvents. The resulting molecular anchoring dilute electrolyte (MADE) is a low concentration electrolyte but retains high ionic conductivity. The electrolyte was well studied by experimental and computational methods, demonstrating the importance of the hydrogen bonding to the solvation structure and electrolyte stability. The Li||NCM811 coin cells and pouch cells were studied, demonstrating the high performance of MADE for high-voltage Li-metal batteries. Moreover, MADE is also able to enhance the thermal stability of Li||NCM811 cells. I think the importance and novelty of this work are high, and I recommend it for publication after revision.

Response: We thank the reviewer for the positive comments on our work. Your professional suggestions and comments are very helpful for us to improve the quality of our manuscript. Our detailed responses are as follows.

1) The authors claimed that the MADE fosters a more flexible SEI layer to accommodate volume changes of Li metal during cycling. Some discussion should be provided to explain this.

Response: We thank the reviewer for the valuable suggestions.

Although SEI with a high modulus is advantageous for suppressing Li dendrite growth, an excessively elevated modulus value would increase the risk of SEI fragmentation (Shen et al. *Adv. Energy Mater.* 2020, 10, 1903645; Chen et al. *Nat Commun.* 2023, 14, 2655). In contrast to the LHCE, the solvation structure in MADEs shows a diminished proportion of anions. This modification serves to balance the decomposition of anions and solvent molecules, thereby promoting the development of a flexible SEI and reducing the susceptibility of the SEI membrane to rupture.

We have incorporated additional discussions into the manuscript as follows.

Page 19: “Although SEI with a high modulus is advantageous for suppressing Li dendrite growth, it is imperative to realize that the substantial Li volume fluctuations during cycling inevitably result in the disruption and subsequent reformation of the SEI (Zhao et al. *Joule*. 2021, 5, 5, 1119-1142). In contrast to the LHCE, the solvation structure in MADEs shows a diminished proportion of anions. This modification serves to balance the decomposition of anions and TTE, thereby promoting the development of a flexible SEI, ultimately reducing the susceptibility of the SEI fragmentation (Fig. 5d).”

2) The MADE consists of a high TTE to DME molar ratio of 3:1. Why is the molar ratio of 3:1 used in this work?

Response: We thank the reviewer for the question. As the proportion of TTE increases, the oxidation stability of the electrolyte increases. Therefore, we chose the 3:1 ratio of TTE to DME to ensure sufficient anchoring of DME, As shown in Fig. R28, the anodic onset potential increases with the TTE ratio. Further increasing the TTE content would undermine the ionic conductivity of the electrolyte.

Additionally, we supplemented the CEs of Li||Cu cells with different ratios of DME to TTE, as shown in Fig. R27. As the content of TTE gradually increases, the Li||Cu CE of the electrolytes first increases and then remains relatively stable. Balancing Li||Cu CE with the salt concentration, DME: TTE=1:3 is an appropriate ratio.

Figure R28. The LSV results for the electrolytes with different TTE ratios using Li||Super P-PVDF cells.

3) More evidence should be provided to confirm the TTE-DME H-bond interactions. For example, the FTIR and ^{17}O NMR can be employed to study the H bonding.

Response: We greatly appreciate the reviewer for the constructive comments. We have conducted additional experiments, such as Infrared, Raman spectroscopy and ^{17}O NMR, to further confirm the hydrogen bonding interactions between TTE and DME.

Firstly, we investigate the intermolecular interactions between TTE and ether via FT-IR in the CCl_4 medium to avoid interference from the existing molecular interactions in the pure solvent. To distinguish the C-H vibrations of $-\text{CF}_2\text{H}$ in TTE with those in DME, we use the deuterated ether (*d*₈-THF) instead of DME (C-D vibrations shift to a lower wavenumber), as it has a very similar C-O-C ether moiety as DME. We apologize for being unable to obtain deuterated DME for a direct comparison due to the export control policy. As shown in Fig. R18, the C-H vibration in TTE shows a discernible blue shift after anchoring with the ether moiety, aligning with the expected blue-shift documented in the literature when $-\text{CF}_2\text{H}$ acts as an H-bond donor (Sessler et al. *J. Am. Chem. Soc.* 2017, 139, 27, 9325–9332). This provides compelling support that $-\text{CF}_2\text{H}$ establishes H-bonding interactions with the oxygen atom in ether.

Secondly, it is known that DME can form hydrogen bonds with aniline, and TTE can form hydrogen bonds with *d*₆-DMSO. From the changes of Raman signals (Fig. R22-R24), we have demonstrated that the impact of TTE on DME is consistent with the impact of aniline on DME, and the effect of DME on TTE aligns with the effect of *d*₆-DMSO on TTE. These results demonstrated that a hydrogen bond exists between TTE and DME.

In addition, we further compared the ^{17}O NMR spectra of mixed solutions with different ratios of DME and TTE (Fig. R29). The results showed that as the content of TTE increased, the ^{17}O signal shifted to a lower chemical shift, which agrees with the previous study that hydrogen bonding induces an upfield shift of ^{17}O -NMR (Reuben et al., *J. Am. Chem. Soc.* 1969, 91, 21, 5725). Meanwhile, there is no apparent shift of ^{17}O -

^{17}O NMR signal for the oxygen atom in TTE. Therefore, the hydrogen bonding between - CF_2H and $\text{O}_{(\text{DME})}$ is further validated.

Figure R29. ^{17}O NMR spectra of mixed solutions with different ratios of DME and TTE.

We have added the above result and the following discussion in the revised manuscript.

“Moreover, with the content of TTE increasing, the oxygen of DME shifted to a lower chemical shift, which agrees with the previous study that hydrogen bonding induces an upfield shift of ^{17}O -NMR (Reuben et al., J. Am. Chem. Soc. 1969, 91, 21, 5725). Meanwhile, there is no apparent shift of ^{17}O -NMR signal for the oxygen atom in TTE. Therefore, the hydrogen bonding between $-\text{CF}_2\text{H}$ and $\text{O}_{(\text{DME})}$ is further validated.”

4) The testing conditions for Li||NMC-811 coin cells and pouch cells should be provided. For example, what is the ratio of anode to cathode? What is the thickness of Li metal anode?

Response: We appreciate the reviewer for this good suggestion. In the coin cells, the cathode (NMC811) loading is 1.5 mAh cm^{-2} , and the Li anode has a thickness of $450 \mu\text{m}$. Celgard 2500 is used as the separator.

The pouch cell parameters are shown in Table R1.

Table R1. Cell parameters of the Li||NMC811 pouch cell.

	Parameter	Value
NMC811 cathode	Capacity	2.1 Ah
	Active material loading	96.5%
	Area capacity (each side)	2.5 mAh cm ⁻²
	Al foil thickness	12 μm
Li anode	total thickness without Cu foil	50 μm
N/P		2.1
Electrolyte	E/C	3 g Ah ⁻¹
Separator	PE	12 μm

5) The full electrochemical stability window of the MADE, showing Li plating and stripping, should be provided.

Response: We appreciate the reviewer for the good suggestion. The anodic stability results can be found in Fig. S6 of our originally submitted manuscript. As shown in Fig. R30, Li||Cu half cells are employed to investigate the cathodic stability and Li plating/stripping behavior in different electrolytes. The MADEs and LHCE show similar reduction and Li plating/stripping behavior during the CV scan. The LHCE shows a more apparent reduction peak at ~1.5 V vs. Li/Li⁺, which is likely corresponds to the pronounced FSI⁻ decomposition. The higher current densities in MADE-2 and MADE-3 are probably due to the higher ionic conductivity of the electrolytes.

**Figure R30.** CV curves of Li plating/stripping in different electrolytes with 5 mV s⁻¹.

6) The Charge-discharge curves of Li||NMC-811 cells upon long-term cycling should be provided to show if there is voltage fade upon cycling.

Response: We appreciate the reviewer for this good suggestion. We have added the long-term cycling charge-discharge curve of Li||NMC811 cells in the supporting information as shown in Fig. R31. MADE-2 enables more stable cycling compared to MADE-1 and MADE-3. As discussed above, a controllable degree of FSI decomposition may have a beneficial effect on the CEI layer, thus suppressing the cathode degradation due to electrolyte side reactions. A very thin CEI layer with LiF, Li₂O, LiSO_x, LiNO_x and etc. (with contribution from FSI decomposition) would have favorable ionic conductivity and passivation ability to shield the reactive electrode surface from labile “free” solvent molecules generated from dynamic motions in the electrolyte.

This result has been supplemented in Figure S14 in the revised manuscript.

Figure R31. Voltage curves of Li||NMC811 in MADE-1, MADE-2, MADE-3 and LHCE.

7) How is the flammability of the MADE?

Response: We thank the reviewer for this question. We conducted flammability tests on the DE, MADEs, and LHCE. The DE is highly flammable and has a very large self-extinguishing time (SET) of 81 s g^{-1} . In contrast, both the MADEs and LHCE cannot be easily ignited due to the large contents of the fluoroether diluent (Fig. R32). The above information, figures and videos have been supplemented in the revised manuscript. And the following description has also been added.

Page 21: “Moreover, the DE is highly flammable and has a very large self-extinguishing time (SET) of 81 s g^{-1} . In contrast, both the MADEs and LHCE cannot be easily ignited due to the large contents of the fluoroether diluent (Supplementary Fig. 40)”

Figure R32. Flammability tests of the DE, MADE-1, MADE-3 and LHCE.

8) What is the reference peak for XPS?

Response: We appreciate the reviewer for this good suggestion. All of the XPS spectra

are referenced to the C 1s peak of C-C/C-H at 284.8 eV. This information has been added to the supporting information.

9) The symmetric Li-Li cell test should be performed to assess the Li metal anode compatibility

Response: We thank the reviewer for this valuable suggestion. Li||Li symmetric cells (0.5 mA cm^{-2} , 1 mAh cm^{-2}) were tested with different electrolytes, as shown in Fig. R33. Compared to the LHCE, symmetrical cells employing MADEs exhibit a notable reduction in overpotentials. However, the Li volume change during plating and stripping would inevitably cause the consumption of the electrolyte, especially the reactive FSI⁻. As a result, the MADEs show a relatively shorter cycle life due to the significantly lower amount of FSI⁻ compared to the LHCE. We expect such problem could be mitigated by manipulating the reactivity of the salt added in the MADE.

Figure R33. Cycling of Li||Li symmetrical cells using MADE-1, MADE-3, and LHCE.

The above result and the following discussion have also been added to the revised manuscript.

“As depicted in Supplementary Fig. 33, when compared to LHCE, symmetrical cells employing MADEs exhibit a notable reduction in overpotential. Nevertheless, the cycling lifespan is observed as LHCE > MADE-3 > MADE-1, primarily attributed to

the lower anion concentration in MADEs. We expect such problem could be mitigated by manipulating the reactivity of the salt used in the future.”

10) The Li cycling CE should also be evaluated using the Aurbach measurement.

Response: We appreciate the reviewer for this valuable suggestion. We followed reviewer's advice and compared the Aurbach Coulombic efficiency (CE) of the studied electrolytes (Fig. R2). Preconditioning of the Cu electrode was performed by depositing and stripping of 2 mAh cm⁻² Li metal at 0.5 mA cm⁻². Then the Li CE was measured after 20 cycles of Li deposition/stripping (1 mAh cm⁻² each cycle, 2 mAh cm⁻² reservoir, deposition at 0.5 mA cm⁻², stripping at 0.5 mA cm⁻²).

Figure R2. The voltage curves of Li||Cu cells with Aurbach's method of DE (a), MADE-1 (b), MADE-2 (c), MADE-3 (d) and LHCE (e).

Moreover, we have included this Aurbach CE in the supplementary materials and added relevant descriptions in the revised version as follows:

Page 17: “The Li CE results from Aurbach’s method exhibit a similar trend (Supplementary Fig. 32). In LHCE, the clustering of anions introduced by a high salt content aggravate the consumption of Li metal by reactions with anion, resulting in only a minor change in the Li CE compared to MADE-3.”

REVIEWER COMMENTS

Reviewer #1 (Remarks to the Author):

The authors have answered all my questions. I support the publication of this work from my side.

Reviewer #2 (Remarks to the Author):

The updated rendition of "Molecular Anchoring of Free Solvents for High-Voltage and High Safety Lithium Metal Batteries" has indeed demonstrated significant improvement compared to its predecessor. The authors have effectively addressed a majority of the queries posed by the reviewers. However, there remain several pivotal questions and issues that necessitate attention before the publication can be justified. Notably, the HOESY spectrum, serving as the primary experimental evidence for hydrogen bonding in this study, demands thorough clarification and resolution.

1. Firstly, while the argument for "relatively strong" intermolecular forces between DME and TTE is persuasive, claiming the existence of a hydrogen bond between these two molecules requires further clarification. An issue arises in Figure 1b, where a significant disparity in the chemical shifts of the two peaks of pure DME is observed when CDCl₃ or DMSO-d₆ is employed as the external reference. The substantial difference prompts the question: what accounts for this variation? It is advisable for the authors to utilize NMR with DMSO-d₆ as the external reference consistently, aligning with established practices in other studies, to enhance methodological coherence and reliability.

2. Considering that the inclusion of DOL in DME induces noteworthy shifts in the DME peaks, and notably, the absence of observable coupling (splitting) between F and H (donor and acceptor) in the 1D proton NMR between TTE and DME, it may be advisable to exclude the 1D-NMR from Fig 1b. The 1D-NMR data might not warrant extensive elaboration. Instead, a more compelling choice could be to reintegrate the HOESY spectrum (Fig S2) into Fig 1b, given its role as significantly stronger evidence in support of the hydrogen bonding argument. This adjustment would enhance the visual representation and reinforce the key experimental findings.

3. Furthermore, more comprehensive explanations regarding the HOESY spectrum are essential, given its pivotal role as the strongest evidence supporting the hydrogen bonding assertions in the paper. The Nuclear Overhauser Effect (NOE) typically manifests between nuclei pairs in close proximity, usually within 0.5 nm. The authors must elucidate why there is a Nuclear Overhauser Effect between F3 (TTE) and both H3 and H4 (DME) in the HOESY spectrum. Additionally, the occurrence of a NOE between F4 (TTE) and both H3 and H4 (DME) necessitates a clearer rationale. While it can be assumed that, based on Figure R15, there might be a NOE between F2 and both H3 and H4 if there is hydrogen bonding between H1 and the O of DME, the authors should provide a detailed explanation for the observed NOE between F4 and both H3 and H4 protons. This becomes particularly intriguing as F4 is already one carbon away from the C-H donor. Special orientations may explain a NOE with either H3 or H4 (Figure R17), but the simultaneous NOE with both H3 and H4 for F4 requires a more explicit elucidation. Similar concerns are raised for F3, which also exhibits a NOE with both H3 and H4, necessitating a clear and detailed explanation for this observation. Clarifying these nuances will strengthen the interpretation of the HOESY spectrum and further support the conclusion of hydrogen bonding.

4. The authors are encouraged to provide further clarification on the utilization of HSQC for detecting hydrogen bonding. Typically employed to identify connections between protons and their respective attaching carbon atoms, as illustrated in Figure R20, HSQC is not conventionally associated with detecting hydrogen bonding. The authors should expound on the rationale behind using HSQC in this context. Furthermore, attention is drawn to the apparent issues within the HSQC spectrum presented in Figure R20. Notably, the observed phases of H2 (TTE), H3, and H4 (DME) appear to be uniform. However, methyl protons (H3) are expected to exhibit a different phase compared to methylene protons (H2 and H4). The authors should address this discrepancy and provide a clear explanation for the phase consistency observed in the HSQC spectrum,

ensuring accurate and reliable interpretation of the experimental data.

Reviewer #3 (Remarks to the Author):

The authors addressed all my questions. I recommended it for publication.

Responses to reviewers' suggestions and comments

We express our gratitude to the reviewers for their invaluable feedback. The reviewers' comments and suggestions have been integrated into the revised manuscript. Additionally, comprehensive responses and elucidations to comments raised by the reviewer have been furnished. The modifications made to the manuscript are highlighted in yellow within this response and in the revised manuscript.

Reviewer #2 (Remarks to the Author):

The updated rendition of "Molecular Anchoring of Free Solvents for High-Voltage and High Safety Lithium Metal Batteries" has indeed demonstrated significant improvement compared to its predecessor. The authors have effectively addressed a majority of the queries posed by the reviewers. However, there remain several pivotal questions and issues that necessitate attention before the publication can be justified. Notably, the HOESY spectrum, serving as the primary experimental evidence for hydrogen bonding in this study, demands thorough clarification and resolution.

Response: We sincerely appreciate the reviewer for providing insightful comments and suggestions. Your recommendations and remarks are invaluable in enhancing the quality of our manuscript. We will address the reviewer's questions point by point.

1. Firstly, while the argument for "relatively strong" intermolecular forces between DME and TTE is persuasive, claiming the existence of a hydrogen bond between these two molecules requires further clarification. An issue arises in Figure 1b, where a significant disparity in the chemical shifts of the two peaks of pure DME is observed when CDCl₃ or DMSO-d₆ is employed as the external reference. The substantial difference prompts the question: what accounts for this variation? It is advisable for the authors to utilize NMR with DMSO-d₆ as the external reference consistently, aligning with established practices in other studies, to enhance methodological coherence and reliability.

Response: We greatly appreciate the reviewer's valuable comment. There are two

dominant factors that influence the chemical shift value of the sample using different external references. The first factor is the chemical shift change of the TMS reference caused by the different magnetic susceptibility of deuterated reagents. The second one is the slight difference in the actual magnetic field experienced by the test sample induced by the varied magnetic susceptibility of deuterated reagents.

When a nuclear spin is in a uniform external magnetic field, magnetic nuclei will undergo rotational motion (precession) around the direction of the external magnetic field. The precession frequency is related to the external magnetic field as follows:

$$\omega = \gamma B_0$$

Where ω is the angular frequency of precession, γ is the gyromagnetic ratio (related to spin nuclei), B_0 is the external magnetic field.

For a typical valid measurement, the basic requirement is to measure the resonance frequencies for the sample and reference under the same magnetic induction value B_0 . This is the case for the conventional internal referencing method, where the sample and the reference are dispersed in a homogeneous liquid within a single test tube. However, the intermolecular interactions between the solvents (DME and TTE in our case) and the deuterated solvents (CDCl_3 and $\text{DMSO-}d_6$) would unavoidably interfere with our study of interactions between DME and TTE. Therefore, we have to rely on the external reference method to study the chemical shifts of DME and TTE by physically separating the sample and the deuterated solvents.

Under the conditions of external reference, although both sample and reference experience the same external magnetic field, their magnetic induction fields are different due to their varied bulk magnetic susceptibility (BMS). Therefore, the measured frequencies must be adjusted to account for the BMS effect. This also applies to the reference TMS protons, whose chemical shift is defined as zero in the actual measured data. Therefore, it is necessary to consider the changes in the chemical shift of TMS in different deuterated reagents, which is essentially related to the BMS of

at 0 ppm (the actual observed chemical shift of TMS protons in DMSO-*d*₆ is 0.54 ppm as indicated in Table R1). Therefore, when using DMSO-*d*₆, the chemical shifts of samples are lower by 0.54 ppm compared to those using CDCl₃, which is close to the difference of 0.51 ppm we found.

On the other hand, under a constant external magnetic field (instrument parameter), due to the different magnetic susceptibilities of deuterated reagents and DME, the actual magnetic fields experienced by DME are varied with different external references. This would also result in a slight change in the obtained chemical shift.

Additionally, to avoid confusion for readers, we have replaced all NMR results (including the ¹H-¹⁹F HOESY) with DMSO-*d*₆ as the external reference (Fig. R1).

Figure R1. Two-dimensional ¹H-¹⁹F NMR spectrum of DME-TTE mixture (1:3 by mole) using DMSO-*d*₆ as the external reference.

2. Considering that the inclusion of DOL in DME induces noteworthy shifts in the DME peaks, and notably, the absence of observable coupling (splitting) between F and H (donor and acceptor) in the 1D proton NMR between TTE and DME, it may be advisable to exclude the 1D-NMR from Fig 1b. The 1D-NMR data might not warrant extensive elaboration. Instead, a more compelling choice could be to reintegrate the

HOESY spectrum (Fig S2) into Fig 1b, given its role as significantly stronger evidence in support of the hydrogen bonding argument. This adjustment would enhance the visual representation and reinforce the key experimental findings.

Response: We sincerely appreciate the reviewer's kind suggestion. We have moved the 1D-NMR data into the supporting information and integrated the HOESY spectrum as well as the FT-IR data into Fig. 1b to better illustrate the interactions between DME and TTE molecules. The revised Fig. 1 is shown below.

Figure 1. Molecular interactions and electrolyte design strategy. **a** Heat flow curves with DME and HCE added into TTE. **b** The two-dimensional ^1H - ^{19}F Heteronuclear Overhauser Effect Spectroscopy (HOESY) for DME-TTE (1:3 by mole). **c** Comparison of C-H vibration signals of TTE before and after mixing with THF- d_8 and DMSO- d_6 in CCl_4 . **d-e** The radical distribution functions and coordination numbers calculated from MD simulations of MADE-1 and LHCE. **f** Schematic diagram of MADE and

HCE/LHCE design strategy.

3. Furthermore, more comprehensive explanations regarding the HOESY spectrum are essential, given its pivotal role as the strongest evidence supporting the hydrogen bonding assertions in the paper. The Nuclear Overhauser Effect (NOE) typically manifests between nuclei pairs in close proximity, usually within 0.5 nm. The authors must elucidate why there is a Nuclear Overhauser Effect between F3 (TTE) and both H3 and H4 (DME) in the HOESY spectrum. Additionally, the occurrence of a NOE between F4 (TTE) and both H3 and H4 (DME) necessitates a clearer rationale. While it can be assumed that, based on Figure R15, there might be a NOE between F2 and both H3 and H4 if there is hydrogen bonding between H1 and the O of DME, the authors should provide a detailed explanation for the observed NOE between F4 and both H3 and H4 protons. This becomes particularly intriguing as F4 is already one carbon away from the C-H donor. Special orientations may explain a NOE with either H3 or H4 (Figure R17), but the simultaneous NOE with both H3 and H4 for F4 requires a more explicit elucidation. Similar concerns are raised for F3, which also exhibits a NOE with both H3 and H4, necessitating a clear and detailed explanation for this observation. Clarifying these nuances will strengthen the interpretation of the HOESY spectrum and further support the conclusion of hydrogen bonding.

Response: We thank the reviewer for the good questions and constructive comments. It should be noted that the DME-TTE dual-molecular complex depicted in our original manuscript was mainly to emphasize the dominating H-bonding intermolecular interactions ($H_{(TTE)}$ and $O_{(DME)}$). Nevertheless, in the electrolyte or a mixture of DME and TTE (1:3 in molar ratio), as illustrated in Figure R2b (a fragment excerpt from one random snapshot of the MD simulation), each molecule could potentially interact with several surrounding molecules, resulting in diverse interaction sites. When further dissecting different DME-TTE coordination configurations in Figure R2b, it can be found that F_3 can simultaneously approach H_3/H_4 ($<3 \text{ \AA}$), and similarly, F_4 can also establish interactions simultaneously with H_3/H_4 (Figure R2c).

We could also observe the evolution of intermolecular interactions when we change the ratio between DME and TTE in the molecular complex. For example, when we introduced additional DME or TTE molecules to the DME-TTE dual-molecular complex, the optimization results indicated the emergence of new interactions in the system, such as F₄-H₄, F₂-H₄, F₃-H₄, F₃-H₃, F₁-H₁ and F₃-H₂. Therefore, the simultaneous NOE with both H₃ and H₄ for F₄ or F₃ is reasonable due to the diverse interactions.

Figure R2. **a** The different molecular configurations of TTE and DME. **b** A fragment excerpt from one random snapshot of the MD simulation. **c** The configurations of the DME-TTE complex extracted from **b**. (Grey: C; White: H; Red: O; blue: F)

Figure R3. **a** The optimized DME-TTE complex. **b** The optimized DME-TTE-DME complex. **c** The optimized TTE-DME-TTE complex.

Despite the diverse interactions shown above, we believe the H-bonding interaction between $H_{(\text{TTE})}$ and $O_{(\text{DME})}$ is the main force that determines the electrolyte solvation structure and the electrochemical performance. This aspect has been discussed in our previous response letter by comparing with hydrofluoroether diluents (HFE-2 and HM) that do not contain the $-\text{CF}_2\text{H}$ moiety.

4. The authors are encouraged to provide further clarification on the utilization of HSQC for detecting hydrogen bonding. Typically employed to identify connections between protons and their respective attaching carbon atoms, as illustrated in Figure R20, HSQC is not conventionally associated with detecting hydrogen bonding. The authors should expound on the rationale behind using HSQC in this context. Furthermore, attention is drawn to the apparent issues within the HSQC spectrum presented in Figure R20. Notably, the observed phases of H2 (TTE), H3, and H4 (DME) appear to be uniform. However, methyl protons (H3) are expected to exhibit a different phase compared to methylene protons (H2 and H4). The authors should address this discrepancy and provide a clear explanation for the phase consistency observed in the HSQC spectrum, ensuring accurate and reliable interpretation of the experimental data.

Response: We thank the reviewer for the careful review and insightful comments. We apologize for accidentally making a mistake in the atom labeling and not using a suitable test sequence to detect the interaction between $H_{(TTE)}$ and $C_{(DME)}$ through H-bonding. The correct $H_{3(DME)}$ and $H_{4(DME)}$ should be as shown in the following Figure R4 (H_3 has a higher chemical shift). The methylene protons (H_2 and H_3) are in the same phase, while the methyl protons (H_4) are in a different phase.

Figure R4. The 1H - ^{13}C HSQC spectrum of the mixture obtained DME and TTE (1:3 by mole).

In addition, it is true that HSQC is best for identifying connections between hydrogen and the attaching carbon atoms. We referred to the previous work for the test (Chattopadhyay et al. *J. Phys. Chem. B.* **2016**, 120, 10679–10685). However, we did not realize that their HMQC/HSQC sequences were specifically modified to detect longer range interactions. Because $H_{(TTE)}$ atoms are spatially close to the $O_{(DME)}$ atoms, it will result in the close proximity between the $H_{(DME)}$ and the $H_{(TTE)}$ atoms. Therefore, 1H - 1H NOSEY can be employed to analyze their through-space interactions (Mann et al. *Mol. Pharmaceutics* **2020**, 17, 2, 622–631; Wohnert et al. *Nucleic Acids Research* **1999**, 15, 3104-3110). As shown in Figure R5, the correlation signals between the $H_{(TTE)}$ and the $H_{(DME)}$ atoms can be clearly observed, and the corresponding spatial

configurations can be explained by Figure R5b. The $H_{\text{(TTE)}}$ is close to the $O_{\text{(DME)}}$ (H_1 - O_{DME} 2.17 Å; H_2 - O_{DME} 2.24 Å), resulting in a spatial proximity of less than 5 Å between the $H_{\text{(DME)}}$ and $C\text{-}H_{\text{(DME)}}$.

Figure R6. **a** The ${}^1\text{H}$ - ${}^1\text{H}$ NOESY spectrum of the DME-TTE mixture (1:3 by mole). **b** The optimized DME-TTE complex configuration.

We appreciate the reviewer's kind comments and suggestions, which are highly beneficial for improving the quality of the manuscript. We hope that the reviewer is satisfied with our response and revision.

REVIEWERS' COMMENTS

Reviewer #2 (Remarks to the Author):

I think the authors have addressed the questions raised by all reviewers. The paper should be ready to publish.